# Cryo-EM structures and functional characterization of murine Slc26a9 reveal mechanism of uncoupled chloride transport

Justin D Walter[†], Marta Sawicka[†], Raimund Dutzler*

Department of Biochemistry, University of Zurich, Zurich, Switzerland

**Abstract** The epithelial anion transporter SLC26A9 contributes to airway surface hydration and gastric acid production. Colocalizing with CFTR, SLC26A9 has been proposed as a target for the treatment of cystic fibrosis. To provide molecular details of its transport mechanism, we present cryo-EM structures and a functional characterization of murine Slc26a9. These structures define the general architecture of eukaryotic SLC26 family members and reveal an unusual mode of oligomerization which relies predominantly on the cytosolic STAS domain. Our data illustrates conformational transitions of Slc26a9, supporting a rapid alternate-access mechanism which mediates uncoupled chloride transport with negligible bicarbonate or sulfate permeability. The characterization of structure-guided mutants illuminates the properties of the ion transport path, including a selective anion binding site located in the center of a mobile module within the transmembrane domain. This study thus provides a structural foundation for the understanding of the entire SLC26 family and potentially facilitates their therapeutic exploitation.
DOI: https://doi.org/10.7554/eLife.46986.001

*For correspondence:
dutzler@bioc.uzh.ch

[†]These authors contributed equally to this work

Competing interests: The authors declare that no competing interests exist.

## Introduction

Solute Carrier 26 family member A9 (SLC26A9) is a membrane-transport protein that in recent years has gained considerable attention due to its intriguing functional properties and promising therapeutic potential (*Balázs and Mall, 2018*). It is found abundantly in the epithelia of lung and stomach (*Chang et al., 2009b*; *Lohi et al., 2002*; *Xu et al., 2005*), where it mediates luminal electrogenic chloride transport and contributes to airway clearance and the production of gastric acid. *Slc26a9*[−/−] knockout mice develop airway and gastrointestinal pathologies (*Anagnostopoulou et al., 2012*; *Liu et al., 2015*; *Xu et al., 2008*), and human genetic polymorphisms in SLC26A9 are correlated with altered severity of airway clearance disorders (*Anagnostopoulou et al., 2012*; *Miller et al., 2015*; *Sun et al., 2012*). Owing to these associations, as well as its co-localization and proposed regulatory interactions with the cystic fibrosis transmembrane conductance regulator (CFTR) (*Avella et al., 2011*; *Bertrand et al., 2017*; *Bertrand et al., 2009*; *Chang et al., 2009a*; *Strug et al., 2016*), SLC26A9 has been proposed as a therapeutic target for treating complications associated with cystic fibrosis and other diseases of the airways and gastrointestinal tract (*Balázs and Mall, 2018*; *Mall and Galietta, 2015*). The protein is a member of the SLC26 family of anion transporters, which play indispensable roles in a wide variety of physiological processes such as hearing, cartilage and bone formation, and epithelial secretion and absorption (*Alper and Sharma, 2013*; *Dorwart et al., 2008b*). The ten functional SLC26 paralogs expressed in mammals are targeted to specific tissues and show a distinct transport mechanism and substrate preference (*Alper and Sharma, 2013*), with the human ortholog SLC26A5 (Prestin) having insignificant transport function and instead operating as voltage-dependent motor protein which is responsible for

**eLife digest** Many processes in the human body are regulated by chloride and other charged particles (known as ions) moving in and out of cells. Each cell is surrounded by a membrane barrier, which prevents ions from entering or exiting. Therefore, to control the levels of ions inside the cell, specific proteins in the membrane act as channels or transporters to provide routes for the ions to pass through the membrane.

Channel proteins form pores that, when open, allow a steady stream of ions to pass through the membrane. Transporter proteins, on the other hand, generally contain a pocket that is only accessible from one side of the membrane. When individual ions enter this pocket the transporter changes shape. This causes the entrance of the pocket to close and then re-open on the other side of the membrane.

Inside the lung, an ion channel known as CFTR provides a route for chloride ions to move out of cells, which helps clear harmful material from the airways. Mutations affecting this protein cause the mucus lining the airways to become very sticky, leading to a severe disease known as cystic fibrosis. CFTR works together with another protein that is also found in the membrane, called SLC26A9. Previous studies have suggested that SLC26A9 also allows chloride ions to pass through the membrane. It was not clear, however, if SLC26A9 operates as an ion channel or a transporter protein, or how the protein is arranged in the membrane.

Now, Walter, Sawicka and Dutzler combined two techniques known as cryo-electron microscopy and patch-clamp electrophysiology to reveal the detailed three-dimensional structure of the mouse version of SLC26A9, which is highly similar to the human form. The experiments found that mouse SLC26A9 proteins form pairs in the membrane referred to as homodimers, which arranged themselves in an unexpected way. Further investigation into the structure of these homodimers suggests that despite having many channel-like properties, SLC26A9 operates as a fast transporter, rather than a true channel.

These findings help us understand the role of SLC26A9 and other similar proteins in the lung and other parts of the body. In the future it may be possible to develop drugs that target SLC26A9 to treat cystic fibrosis and other severe lung diseases.

DOI: https://doi.org/10.7554/eLife.46986.002

electromotility in cochlear outer hair cells (*Dallos and Fakler, 2002*). The transport mode of SLC26A9 is controversial, with some reports providing evidence of chloride/bicarbonate exchange (*Chang et al., 2009b*; *Xu et al., 2005*), yet others claiming that SLC26A9 has negligible bicarbonate permeability and instead confers an uncoupled, constitutively-active, channel-like Cl⁻ conductance (*Bertrand et al., 2009*; *Dorwart et al., 2007*; *Loriol et al., 2008*). Although less characterized, uncoupled Cl⁻ transport has also been reported for SLC26A7 (*Kim et al., 2005*) and SLC26A11 (*Rahmati et al., 2013*) whereas evidence for mixed uncoupled and coupled exchange properties was shown for SLC26A3 and SLC26A6 (*Ohana et al., 2011*; *Shcheynikov et al., 2006*).

No high-resolution structures have yet been reported for SLC26A9 or any other eukaryotic SLC26 family member, and detailed structural information about a full-length SLC26 protein is limited to a single crystal structure of a bacterial fumarate transporter (SLC26Dg) (*Geertsma et al., 2015*). The monomeric SLC26Dg structure provided valuable information about the basic elements of the SLC26 architecture consisting of a transport domain of 14 membrane-inserted α-helices, and a C-terminal cytosolic STAS domain of unknown function. However, SLC26 transporters assemble into functional oligomers (*Compton et al., 2014*; *Detro-Dassen et al., 2008*; *Zheng et al., 2006*), and the quaternary arrangement of an oligomeric eukaryotic SLC26 transporter has remained unknown. Moreover, structural knowledge of mammalian paralogs is indispensable for the comprehension of the molecular basis for the mechanistic diversity of the SLC26 family.

Here, we present cryo-electron microscopy (cryo-EM) structures of dimeric murine Slc26a9 accompanied by its detailed functional characterization. The structures of an inward-facing conformation as well as a potentially intermediate state of the transport cycle reveal a protein architecture that is likely representative for all mammalian SLC26 paralogs, including the unusual mode of dimerization in which nearly all the interactions between the two subunits are mediated by the STAS

domains. In combination with electrophysiological experiments, our work demonstrates how Slc26a9 mediates the uncoupled transport of Cl⁻ via a rapid alternate-access mechanism.

## Results

### Functional characterization of Slc26a9

Upon transfection of HEK293 cells with DNA encoding full-length murine Slc26a9, high expression is observed, but the protein is poorly detergent-extractable and exhibits a tendency to aggregate. Therefore, we designed a minimal construct in which two predicted intrinsically disordered regions (IDRs) were removed from the C-terminal part of the protein (*Figure 1—figure supplement 1*), a strategy which allowed crystallization of the isolated STAS domain of Prestin (*Lolli et al., 2016*; *Pasqualetto et al., 2010*). One excised region consists of residues P558-V660 and encompasses a hypervariable IDR which was previously termed the STAS intervening sequence (IVS) (*Dorwart et al., 2008a*; *Pasqualetto et al., 2010*). The second deleted region comprises the final 44 residues of the C-terminus (CT, residues P745-L790) subsequent to the predicted STAS domain fold and contains a terminal PDZ motif which commonly interacts with multi-domain scaffolding proteins (*Kim and Sheng, 2004*). The resulting dual ΔIVS/ΔCT truncation construct has been designated Slc26a9$^T$.

A striking observation from HEK293 cells expressing fluorescently-tagged Slc26a9$^T$ is that the truncations have apparently resulted in greater surface expression (*Figure 1—figure supplement 2A*). This is also observed for a construct only lacking the IVS (SLC26A9$^{ΔIVS}$), but not for a construct solely lacking the CT (SLC26A9$^{ΔCT}$), thus suggesting that the IVS might be responsible for the retention of the protein in intracellular membranes (*Figure 1—figure supplement 2A*). Relative to the full-length protein, Slc26a9$^T$ exhibits several additional favorable qualities, including more efficient detergent extraction and the capacity to be purified to homogeneity in a dimeric state (*Figure 1—figure supplement 2B–C*). To assess whether Slc26a9$^T$ retains transport activity, the purified protein was reconstituted into lipid vesicles and tested for electrogenic Cl⁻ transport using a fluorometric assay (*Kane Dickson et al., 2014*). In these experiments, we observed robust transport activity that can be diminished in a dose-dependent manner by a known inhibitor of Slc26a9, implying that neither the removal of the IDRs from the STAS domain nor the purification procedure has compromised the function of Slc26a9$^T$ (*Figure 1A*, *Figure 1—figure supplement 2D*). Observation of Cl⁻ transport by reconstituted Slc26a9$^T$ also demonstrates that the protein does not require the presence of endogenous cellular factors for its activity.

For a thorough functional characterization of the transport activity, we recorded currents from HEK293T cells transfected with DNA encoding either full-length Slc26a9 or Slc26a9$^T$. In the whole-cell configuration, both constructs yield constitutively active, anion-selective currents with linear current-voltage (I-V) relationships, as reported previously for human SLC26A9 (*Bertrand et al., 2009*; *Chang et al., 2009b*; *Dorwart et al., 2007*; *Loriol et al., 2008*) (*Figure 1—figure supplement 2E–H*). As expected from the higher surface expression, *Slc26a9$^T$*-transfected cells show raw currents which are, on average, at least four-fold larger than currents associated with full-length Slc26a9 (*Figure 1—figure supplement 2E–H*). Moreover, whereas we could not reliably measure macroscopic currents in inside-out patches from full-length *Slc26a9*-transfected cells, robust activity was observed for Slc26a9$^T$ (*Figure 1B*). As for whole-cell experiments, these currents are constitutive and exhibit a linear I-V relationship. Contrary to a previous study (*Chang et al., 2009b*), we did not observe any discrete single-channel transitions in any of our recordings. Since excised patches allow better control of intra- and extracellular conditions, we used this method to characterize functional properties for the remainder of our investigations.

To assess the relative anion/cation selectivity, we measured I-V relationships for inside-out patches of Slc26a9$^T$-expressing cells that were subjected to different NaCl gradients. In all cases, the currents reverse very close to the Nernst potential of Cl⁻, implying that Cl⁻ is the only transported charge in the system and excluding obligate coupling to other ions (*Figure 1C*; *Figure 1—figure supplement 3A*). Next, we evaluated the relationship between substrate concentration and ion flux and found a saturation of currents at high Cl⁻ concentrations (with a $K_m$ of 30 mM, *Figure 1D*), consistent with a weak interaction of the protein with its transported ion. Although the high Cl⁻ selectivity, fast kinetics, and weak substrate interaction would all be consistent with a channel-like mechanism, they do not exclude transport by an alternate-exchange mechanism, since a similar high

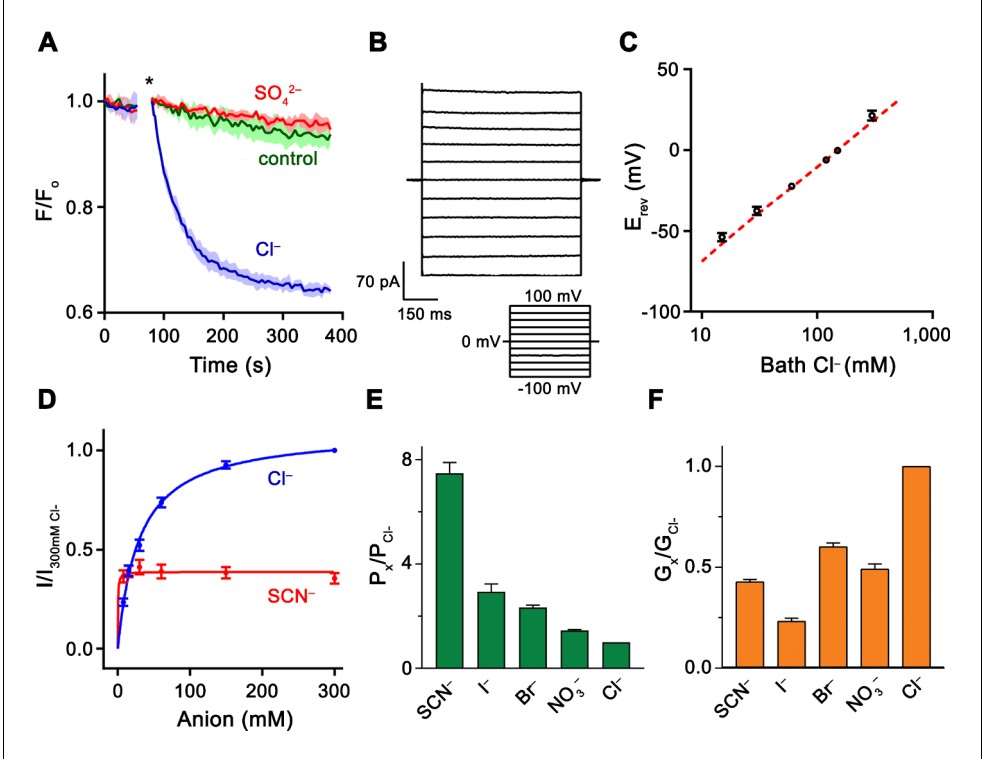

**Figure 1.** Functional properties of Slc26a9. (**A**) Cl⁻ transport of Slc26a9$^T$ reconstituted into proteoliposomes, monitored by the fluorescence change of the pH gradient-sensitive fluorophore ACMA. (*) Indicates addition of the H⁺ ionophore CCCP, which allows counterion movement and electrogenic Cl⁻ transport to proceed. Green trace (control) corresponds to liposomes not containing any protein in the presence of an inward Cl⁻ gradient, red trace ($SO_4^{2-}$) to proteoliposomes exposed to external $SO_4^{2-}$ and blue trace (Cl⁻) to proteoliposomes exposed to external Cl⁻. Traces show mean of three technical replicates. (**B**) Representative current trace recorded from excised patches from HEK293T cells expressing Slc26a9$^T$ at symmetrical 150 mM Cl⁻. Inset shows voltage protocol. (**C**) Reversal potentials at 150 mM extracellular NaCl and varying intracellular NaCl concentrations recorded from inside-out patches ($P_{Na+}/P_{Cl-}$=0.02). Red line corresponds to the Cl⁻ Nernst potential. Presented data is the mean of 5 biological replicates, except for 300 mM, for which n = 3. (**D**) Conductance–concentration relationships of anion transport across Slc26a9$^T$. Data were recorded at −100 mV from inside-out patches with a pipette solution containing 7.5 mM Cl⁻ and varying intracellular Cl⁻ (blue) or SCN⁻ (red) concentrations. $K_m$ (Cl⁻)=29.5 mM; $K_m$ (SCN⁻)=0.5 mM; n = 5, errors are s.e.m.. (**E**) Permeabilities and (**F**) conductance ratios (recorded at −100 mV) were obtained from bi-ionic substitution experiments. Biological replicates for $P_x/P_{Cl}$ and $G_x/G_{Cl}$: Cl⁻, n = 10; Br⁻, n = 10; $NO_3^-$, n = 9; SCN⁻, n = 8; I⁻, n = 6. C-E, Errors are s.e.m.

DOI: https://doi.org/10.7554/eLife.46986.003

The following figure supplements are available for figure 1:

**Figure supplement 1.** Sequence alignment and topology.
DOI: https://doi.org/10.7554/eLife.46986.004

**Figure supplement 2.** Expression, purification and functional characterization of Slc26a9$^T$.
DOI: https://doi.org/10.7554/eLife.46986.005

**Figure supplement 3.** Functional properties of Slc26a9$^T$.
DOI: https://doi.org/10.7554/eLife.46986.006

$K_m$ for Cl⁻ (i.e. ~46 mM) was described for the rapid Cl⁻/$HCO_3^-$ exchanger SLC4A1/Band 3 (**Liu et al., 1996**).

To characterize the ability of Slc26a9$^T$ to discriminate between different anions, we pursued bi-ionic substitution experiments. The observed permeability sequence (SCN⁻>I⁻>Br⁻ > $NO_3^-$>Cl⁻) follows a lyotropic series (**Wright and Diamond, 1977**), which favors more easily dehydrated anions (**Figure 1E**; **Figure 1—figure supplement 3B,C**). In contrast, we measured negligible permeabilities for the anions F⁻, $HCO_3^-$, and $SO_4^{2-}$ (**Figure 1—figure supplement 3D–F**). The permeability

sequence agrees with previous measurements of human SLC26A9 derived from oocyte voltage-clamp experiments (*Dorwart et al., 2007*; *Loriol et al., 2008*). However, the absence of $HCO_3^-$ permeability is in contrast to some reports showing evidence of $Cl^-/HCO_3^-$ exchange by Slc26a9 which were based on measurements of intracellular pH changes in response to removal of external $Cl^-$ in the presence of bath $HCO_3^-$ (*Chang et al., 2009b*; *Xu et al., 2005*). While we cannot fully account for this discrepancy we note that all reports of electrogenic transport indicate negligible movement of $HCO_3^-$. To assess relative transport rates, macroscopic conductance ratios ($G_x/G_{Cl}$) of permeable anions revealed a conductivity sequence of $Cl^->Br^- > NO_3^-\sim SCN^->I^-$ (*Figure 1F*; *Figure 1—figure supplement 3B,C*). Notably, the conductivity pattern deviates from the permeability sequence, with $Cl^-$ being considerably more conductive than any other tested anion. The high relative permeability of lyotropic anions shows that they face a lower energy barrier for dehydration and entry into the transport pathway compared with $Cl^-$, while their lower relative conductance implies that they also encounter a deeper energy well, which in turn would decrease their overall turnover rate. These properties are also reflected in the lower $K_m$ and $V_{max}$ of $SCN^-$ compared to $Cl^-$ (*Figure 1D*) and suggest that Slc26a9 has evolved as an efficient uncoupled $Cl^-$ transporter.

## Slc26a9 structure

To reveal how the functional features of Slc26a9 as a $Cl^-$ transport protein with channel-like properties relate to the protein architecture, we determined its structure by cryo-EM (*Figure 2—figure supplements 1–3*). A 3.96 Å dataset of Slc26a9[T] purified in the synthetic digitonin analogue glyco-diosgenin (GDN) was of high quality and allowed the unambiguous interpretation of the cryo-EM

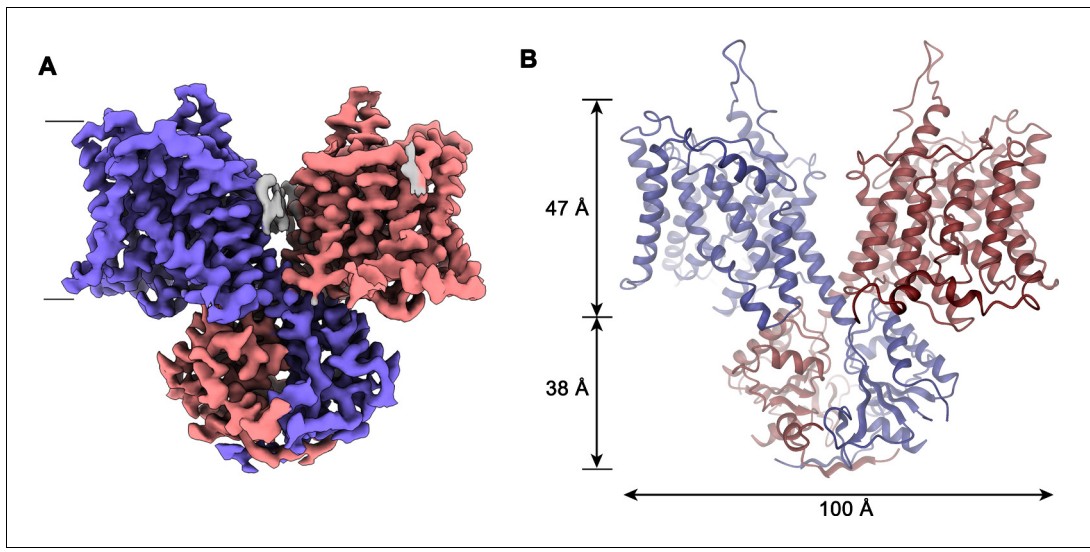

**Figure 2.** Slc26a9[T] structure. (**A**) Cryo-EM density of Slc26a9[T] in the detergent GDN at 3.96 Å contoured at 5σ. Density corresponding to distinct subunits in the dimeric protein is colored in blue and red respectively. Residual density outside the protein region probably corresponding to detergent or lipids is colored in gray. The view is from within the membrane; the boundary of the bilayer is indicated. (**B**) Ribbon representation of Slc26a9[T] with subunits colored in blue and red. Protein dimensions are indicated. View is as in (**A**).
DOI: https://doi.org/10.7554/eLife.46986.007

The following figure supplements are available for figure 2:

**Figure supplement 1.** Cryo-EM reconstruction of Slc26a9[T] in detergent at 3.96 Å.
DOI: https://doi.org/10.7554/eLife.46986.008

**Figure supplement 2.** Cryo-EM density of the Slc26a9[T] structure in detergent.
DOI: https://doi.org/10.7554/eLife.46986.009

**Figure supplement 3.** Cryo-EM reconstruction of Slc26a9[T] in nanodiscs at 7.8 Å.
DOI: https://doi.org/10.7554/eLife.46986.010

**Figure supplement 4.** Cryo-EM density of the Slc26a9[T] structure in nanodiscs.
DOI: https://doi.org/10.7554/eLife.46986.011

density by an atomic model (**Figure 2**; **Figure 2—figure supplements 1** and **2**; **Table 1**; **Video 1**). To exclude a possible bias from a detergent environment, we have in parallel reconstituted the purified protein into lipid nanodiscs and determined its structure at low (7.77 Å) resolution (**Figure 2— figure supplements 3** and **4**; **Video 2**). Both datasets display similar general features of the protein with respect to the organization of subunits and their oligomeric arrangement, although the nanodisc structure reveals localized rigid-body movements indicative of a different functional conformation.

The structure of $Slc26a9^T$ is displayed in **Figure 2B**. The protein is a dimer of elongated shape that is approximately 100 Å long and 50 Å wide (**Video 3**). Next to the membrane-inserted transport domains, two prominent structural features, which are striking in the cryo-EM density, extend

**Table 1.** Cryo-EM data collection, refinement and validation statistics.

| | Dataset 1 $Slc26a9^T$ in detergent (EMD-4997) (PDB 6RTC) | Dataset 2 $Slc26a9^T$ in nanodiscs (EMD-4998) (PDB 6RTF) |
|---|---|---|
| **Data collection and processing** | | |
| Microscope | FEI Titan Krios | FEI Tecnai $G^2$ Polara |
| Camera | Gatan K2 Summit + GIF | Gatan K2 Summit + GIF |
| Magnification | 6,511 | 37,313 |
| Voltage (kV) | 300 | 300 |
| Electron exposure (e–/Å$^2$) | 70 | 60 |
| Defocus range (μm) | −0.5 to −3.0 | −0.5 to −3.0 |
| Pixel size (Å)[*] | 1.075 (0.5375) | 1.34 |
| Symmetry imposed | C2 | C2 |
| Initial particle images (no.) | 416,164 | 711,032 |
| Final particle images (no.) | 112,930 | 17,442 |
| Map resolution (Å) FSC threshold 0.143 | 3.96 | 7.77 |
| Map resolution range (Å) | 3.0–4.2 | 6.0–10.0 |
| **Refinement** | | |
| Model resolution (Å) FSC threshold 0.5 | 4.0 | 8.0 |
| Model resolution range (Å) | 118–3.96 | |
| Map sharpening b-factor (Å$^2$) | −205 | −512 |
| Model composition Non-hydrogen atoms Protein residues | 9570 1242 | 9570 1242 |
| $B$ factors (Å$^2$) Protein | 73 | 73 |
| Refmac FSC$_{avg}$/Rfactor | 0.851/0.351 | |
| R.m.s. deviations Bond lengths (Å) Bond angles (°) | 0.005 0.888 | 0.006 1.195 |
| Validation MolProbity score Clashscore Poor rotamers (%) | 1.10 1.19 0 | 1.07 0.78 0 |
| Ramachandran plot Favored (%) Allowed (%) Disallowed (%) | 96.10 3.90 0 | 95.28 4.72 0 |

[*]Values in parentheses indicate the pixel size in super-resolution.

DOI: https://doi.org/10.7554/eLife.46986.012

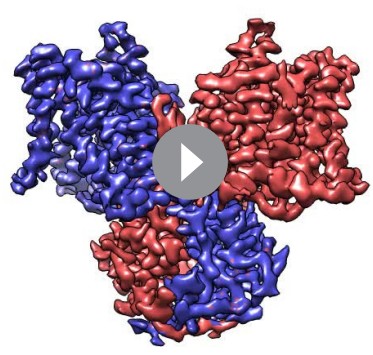

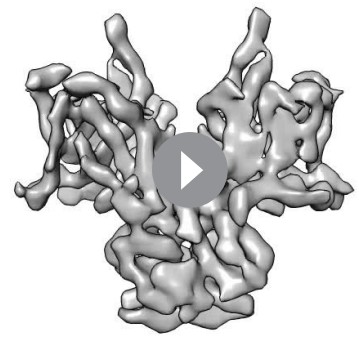

**Video 1.** Cryo-EM density map of the dimeric Slc26a9[T]. Shown is the cryo-EM map of the protein in detergent with the refined model of the inward-facing state superimposed.
DOI: https://doi.org/10.7554/eLife.46986.013

**Video 2.** Cryo-EM density map of the dimeric Slc26a9[T] in nanodiscs. Shown is the cryo-EM map of the protein in nanodiscs with the modeled structure of the intermediate state superimposed.
DOI: https://doi.org/10.7554/eLife.46986.014

beyond the lipid bilayer and flank the membrane-spanning regions. The interacting domain-swapped STAS domains, appearing as a knob centered on one side of Slc26a9[T] particles, protrude 38 Å into the cytosol, while a spiky loop on each subunit projects towards the extracellular solution (*Figure 2*; *Figure 2—figure supplements 1–4*). The interactions between both subunits of the dimeric protein are highly unusual since, unlike other proteins with a similar fold, they only exhibit minimal contacts between the membrane–inserted domains at the intracellular end of the last transmembrane helix (*Figure 2*). Of the 8'114 Å$^2$ of the combined molecular surface buried in the dimer, only 11% can be attributed to interactions directly between the transmembrane domains (TMDs). Conversely, 16% of the buried surface is contributed by mutual contacts between STAS domains whereas the remaining interactions are mutually formed between the swapped STAS domain and the transmembrane part of the opposing subunit. Another remarkable feature of the dimer interface concerns the binding of the extended N-terminus to the STAS domain, which appears to contribute to dimer stability (*Figure 3—figure supplement 1*).

## The STAS domain as interaction platform

In the Slc26a9[T] dimer, the cytoplasmic STAS domains act as a platform for interactions between subunits, thus explaining the deleterious transport phenotype observed upon mutations or its removal in different family members (*Babu et al., 2010*; *Dorwart et al., 2008a*; *Geertsma et al., 2015*; *Sharma et al., 2011*) (*Figure 3*, *Video 3*). This cytoplasmic unit closely resembles known STAS domain structures from eukaryotic SLC26A5 (Prestin) (*Lolli et al., 2016*; *Pasqualetto et al., 2010*) and the bacterial homolog SLC26Dg (*Geertsma et al., 2015*) (*Figure 3—figure supplement 2A,B*). It consists of a six-stranded mixed β-sheet core, which is decorated with four interspersed α-helices (*Figure 1—figure supplement 1* and *Figure 3—figure supplement 2A*). Several of the loop regions connecting β-strands with their adjoining α-helices (i.e. the β3-Cα1, β4-Cα2, β5-Cα3 and the Cα3-β6 loops) make contacts between the STAS

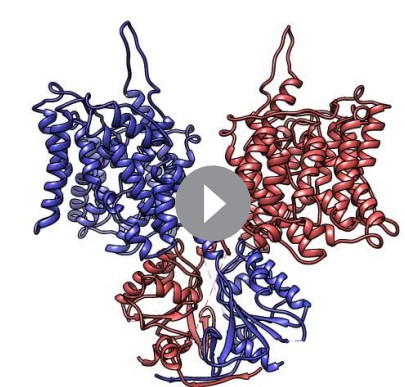

**Video 3.** The structure of Slc26a9[T]. Shown are unique features of a mammalian SLC26 transporter. The oligomerization interface is minimal between the transmembrane domains and is predominantly mediated by the swapped STAS domains. The view of the transmembrane domain only shows two independent modules: core and gate.
DOI: https://doi.org/10.7554/eLife.46986.015

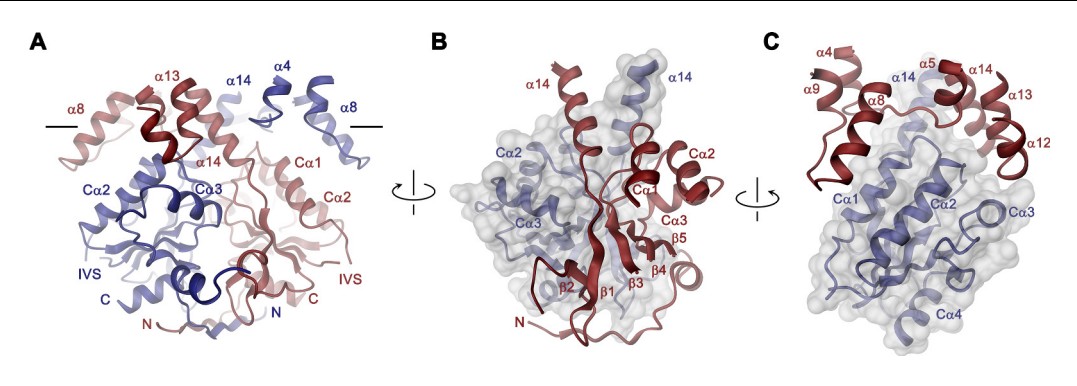

**Figure 3.** STAS domain and dimer interface. (**A**) Ribbon representation of the STAS domain dimer and interacting parts of the N-terminus and the TMD. The view is as in *Figure 2B*. (**B**) Elements of mutual STAS domain interactions and contacts with the N-terminus. (**C**) STAS domain-TMD interactions. B and C, Molecular surface of the STAS domain and the C-terminal part of α14 is shown superimposed on one subunit (blue), segments of the interacting subunit are shown in red. The relationship between views is indicated. A-C, The intracellular membrane boundary is indicated.

DOI: https://doi.org/10.7554/eLife.46986.016

The following figure supplements are available for figure 3:

**Figure supplement 1.** Effect of N-terminal truncation of Slc26a9$^T$ on function and stability.

DOI: https://doi.org/10.7554/eLife.46986.017

**Figure supplement 2.** Structural features of the Slc26a9 STAS domain.

DOI: https://doi.org/10.7554/eLife.46986.018

domains and the TMD of the opposing subunit at parts that are close to the membrane (at the α4-α5, α8-α9, α12-α13 loops and the terminal helix α14) (*Figure 3*). The strong conservation of residues at the contact region with the TMD points towards the preservation of this interface among family members (*Figure 1—figure supplement 1A* and *Figure 3—figure supplement 2C*). At the bottom part of the STAS dimer, interactions with the N-terminus are mediated by the β1-β2, β2-β3 and β6-Cα4 loops (*Figure 3—figure supplement 2D*). Between both regions, on the side of the STAS dimer, a negatively charged groove at the dimer interface could potentially interact with cellular factors (*Figure 3—figure supplement 2E,F*). Finally, in the observed dimeric architecture of the STAS domain, the two unstructured regions removed in the Slc26a9$^T$ construct would be located opposite to the dimer-interface and thus be available for interactions with accessory proteins (*Figure 3—figure supplement 2G*).

## The transmembrane domain

The overall architecture of the TMD conforms to the 7 + 7 inverted repeat topology (*Figure 1—figure supplement 1* and *Figure 4—figure supplement 1*), which was previously observed in the structure of the prokaryotic homologue SLC26Dg (*Geertsma et al., 2015*) and in members of the distantly related SLC4 (*Arakawa et al., 2015*; *Huynh et al., 2018*) and SLC23 (*Alguel et al., 2016*; *Lu et al., 2011*) families. Consistent with the known features of this arrangement, the Slc26a9$^T$ TMD is composed of 14 membrane-inserted α-helices, with α8–14 representing a similar yet inverted organization to α1–7 (*Figure 4—figure supplement 1*). The TMD is segregated into two subdomains, a convex core module (consisting of α1–4, and 9–11) carrying a putative ion-binding site, and a concave gate module (consisting of α5–7 and 12–14) (*Figure 4*, *Video 3*). Both modules constitute independent structural units that interact via an extended interface (*Figure 4A*). Although displaying channel-like functional properties, the structure of the TMD of Slc26a9$^T$ shows hallmarks of a transporter. The striking degree of similarity revealed from a superposition with its equivalent unit of SLC26Dg (RMSD of 2.0 Å) emphasizes the high conservation within the family and suggests that both structures display equivalent conformations (*Figure 5A–C*). The large cavity between gate and core modules is accessible from the intracellular side but does not extend into a contiguous pore region towards the extracellular side, suggesting that the Slc26a9$^T$ structure represents an inward-open state of the protein that facilitates transport by an alternate-access mechanism. This assumption is further supported by the comparison of TMDs of Slc26a9$^T$ from cryo-EM maps acquired either

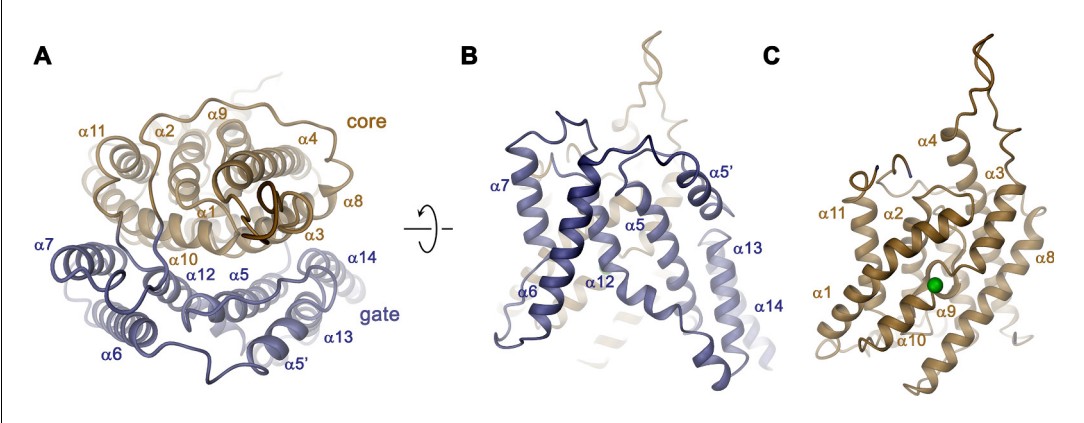

**Figure 4.** Transmembrane domain. (**A**) Ribbon representation of the TMD of a single subunit of Slc26a9$^T$ viewed from the extracellular side. (**B**) View of the gate and (**C**) the core module from within the membrane. Green sphere indicates predicted position of bound Cl$^-$.
DOI: https://doi.org/10.7554/eLife.46986.019

The following figure supplement is available for figure 4:

**Figure supplement 1.** Structural features of the TMD.
DOI: https://doi.org/10.7554/eLife.46986.020

in detergent micelles or lipid nanodiscs (*Figure 2—figure supplement 4C–E*). The nanodisc TMD structure differs in the conformation of the core module, which has rotated as rigid body by approximately 14° and moved by 4 Å towards the extracellular side, whereas the remainder of the protein appears largely unaltered. This conformation thus likely shows an intermediate state of the protein on its transition towards an outward-facing state (*Figure 5D*; *Video 4*).

Despite the general similarity between the TMDs of Slc26a9$^T$ and SLC26Dg, there are several structural features that distinguish the eukaryotic transporter form its prokaryotic homologue. First, a loop-helix structure connecting α5-α6 (α5′) fills a depression in the structure created by the short α13–α14 turn, which would not otherwise reach the boundary of the outer leaflet of an undistorted bilayer (*Figures 4B* and *5B*). A second distinguishing feature of Slc26a9$^T$ concerns the loop region connecting α3 and the extended α4, which protrudes into the extracellular space and which contains two predicted glycosylation sites (*Li et al., 2014*) near the tip of the spiky extension (*Figures 4C* and *5C*). Finally, the extended N-terminus of Slc26a9$^T$, preceding α1, is significantly longer than the equivalent region in SLC26Dg (*Figure 5C*). In the mammalian transporter, the N-terminus is partly mobile and contains extended regions with interspersed secondary structure elements. Whereas the initial part of the N-terminus engages in the earlier described interaction with the STAS domains, a mobile linker bridges this region to a structured part preceding α1, which forms scattered helical segments peripherally contacting the inner membrane leaflet and containing several basic residues (*Figure 5C*, bottom inset). Due to the high sequence conservation between paralogs (*Figure 1—figure supplement 1A*), all three structural features are likely preserved in mammalian SLC26 transporters.

## Structural and functional properties of the ion transport pathway

In the observed inward-facing conformation of the TMD, transported anions enter the protein to access their binding site via a wide aqueous vestibule, presumably attracted by the positive electrostatic potential that is conferred by an excess of basic amino acids at its intracellular entrance (*Figure 6A,B*). The importance of these basic residues for transport is illustrated in the comparison of conductance properties of Slc26a9$^T$ and constructs wherein selected positively charged residues at the intracellular entrance were mutated to glutamate. For wild-type (WT) Slc26a9$^T$, the linear I-V relationship is compatible with symmetric rate-limiting barriers on the transport cycle in the absence of applied voltage (*Lam and Dutzler, 2018*; *Paulino et al., 2017*). This linear relation is minimally perturbed upon introducing the mutations R205E or K358E, which are positioned at the outermost

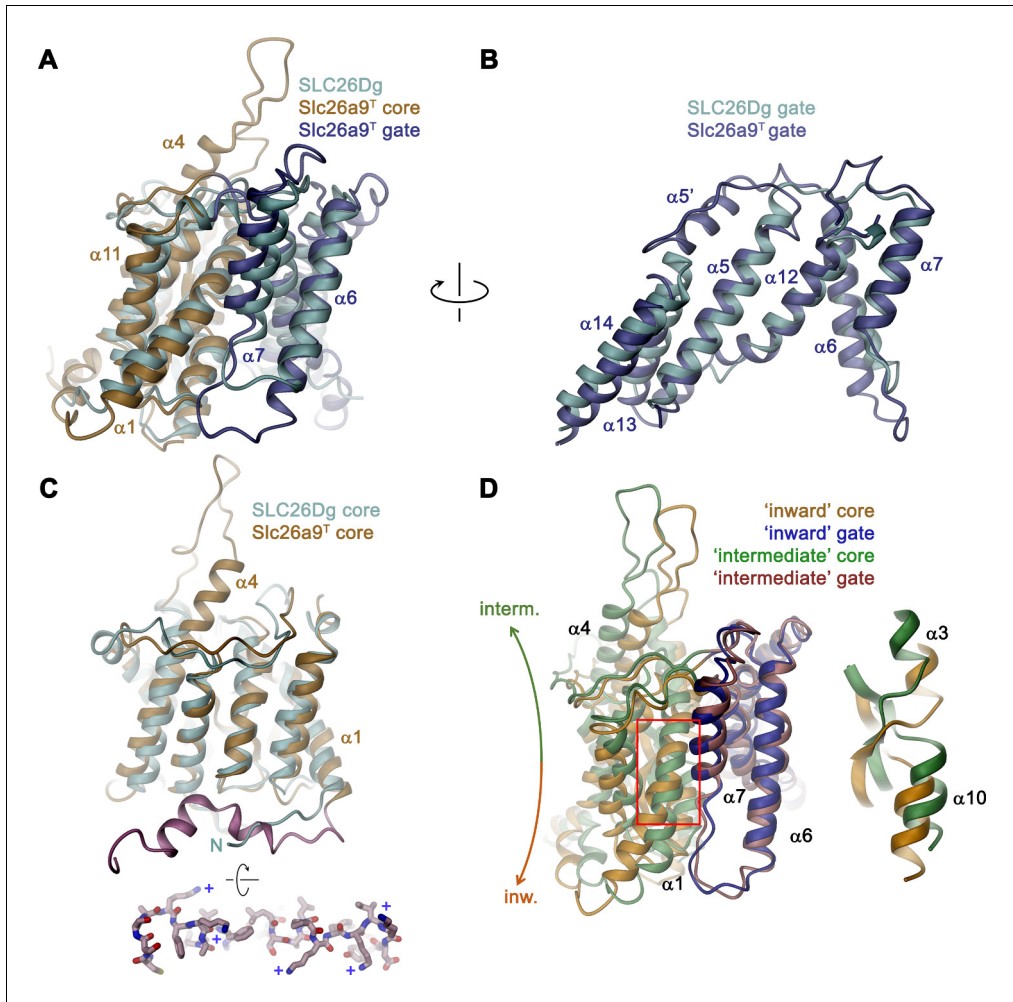

**Figure 5.** Comparison of the TMD. (**A**) Structural superposition of the TMDs of Slc26a9$^T$ (beige and blue) and SLC26Dg (PDBID: 5DA0, cyan). (**B**) Superposition of the gate domains and (**C**) core domains of Slc26a9$^T$ and SLC26Dg. N-terminal region of Slc26a9$^T$ preceding α1 is colored in pink and displayed as sticks from an intracellular perspective below. The location of basic residues is indicated (+). (**D**) Superposition of the TMD of detergent-solubilized Slc26a9$^T$, showing an inward-facing state onto the TMD of lipid nanodisc-reconstituted Slc26a9$^T$, indicating an intermediate state on its transition towards the outside. The movement of the core domains is indicated. Inset (right) shows blow-up of the region around the Cl$^-$-binding site (indicated by red box). The core and gate modules of the intermediate conformation of Slc26a9$^T$ determined in nanodiscs are colored in green and pink, respectively.

DOI: https://doi.org/10.7554/eLife.46986.021

regions of the intracellular vestibule (*Figure 6A*; *Figure 6—figure supplement 1A*). In contrast, mutations K270E and K443E, located deeper in the intracellular cavity and close to the presumed anion binding site, both altered the electrostatics and displayed considerable outward rectification (*Figure 6A–C*). This behavior indicates a relative increase of the inward barrier by either impeding the access of anions from the intracellular side or alternatively by hampering conformational transitions of the protein. Complimentary to the intracellular charge-reversal mutations, we find a reciprocal effect leading to inward rectification upon mutation of extracellular basic residues K221 and K431 that are predicted to line the aqueous cavity to the binding site in an outward-facing conformation (*Figure 6—figure supplement 1B,C*).

Although in our cryo-EM reconstructions we did not observe any density attributable to bound Cl$^-$, its absence is concordant with its measured low apparent affinity. We infer an anion binding site to be located in a pocket of the core domain between the N-termini of the membrane-inserted

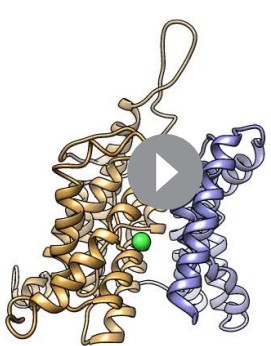

**Video 4.** Conformational transition from the inward-facing to the intermediate state. Ribbon representation of the transmembrane domain as a morph between the inward-facing and intermediate states. The structure is viewed from the peripheral. Residues 276 to 312 are removed for clarity. The putative trajectory of the chloride ion is also shown.

DOI: https://doi.org/10.7554/eLife.46986.022

partial helices α3 and α10 (*Figures 4C* and *7A*; *Video 5*). The equivalent location was proposed as a substrate-binding site for the fumarate-proton symporter SLC26Dg (*Chang et al., 2019a*) and the chloride-bicarbonate transporter SLC4A1/Band 3 (*Arakawa et al., 2015*) and confirmed for the SLC23 transporters UraA (*Lu et al., 2011*) and UapA (*Alguel et al., 2016*). This site would be ideally positioned at the apex of the inward-open cavity, with its dual pseudo-symmetrically related helix dipoles arranged in a manner frequently observed for binding sites of anion transporters (*Dutzler et al., 2002*; *Screpanti and Hunte, 2007*) (*Figures 4C* and *7A*). Compared to SLC26Dg, the pocket of Slc26a9$^T$ is considerably smaller, as a consequence of the conformation of F128 projecting its side-chain into the pocket, thus creating a site of appropriate size for a Cl$^-$ ion (*Figure 7A*). In this site, several residues could plausibly interact with the transported substrates. Besides F128, these include the nonpolar residues F92 and L391, all contributing to the partly hydrophobic character frequently found in anion binding sites which likely account for the observed lyotropic permeability sequence of anions (*Figure 1E*) (*Dani et al., 1983*; *Dutzler et al., 2002*). Dehydrated anions could be stabilized by side-chain interactions with Q88, S392, and possibly T127, as well as main-chain interactions at the N-terminus of α10, which together form a cradle to coordinate the desolvated ion (*Figure 7A*). To probe the functional importance of potential binding-site residues for anion interactions, we individually mutated them to alanine and used inside-out

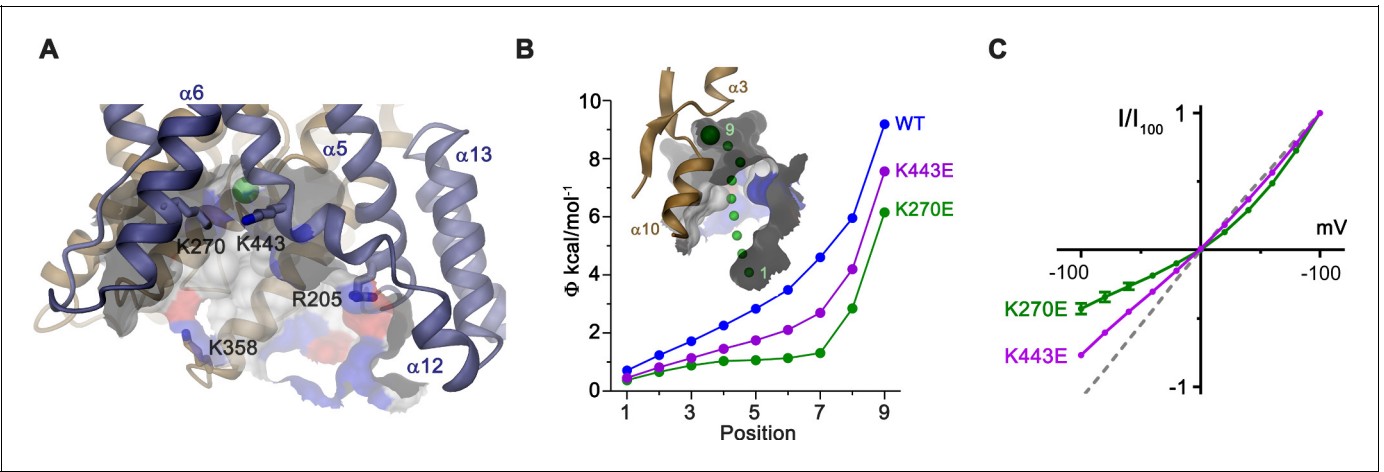

**Figure 6.** Electrostatic properties of the intracellular cavity. (A) View of the intracellular cavity leading to the Cl$^-$-binding site. Section of the molecular surface is shown with contact region of acidic residues colored in red and basic residues in blue. (B) Electrostatic potential within the aqueous cavity leading to the Cl$^-$-binding site of WT and charge reversing mutants of basic residues facing the cavity. Inset shows aqueous access path of ions. Positions sampling the electrostatic potential are shown as green spheres. A, B, Bound Cl$^-$ is shown as a large green sphere. (C) I-V relationships of charge-reversing mutants K270E and K443E facing the aqueous cavity (K270E, n = 3; K443E, n = 4). Data was recorded from excised patches in symmetric 150 mM Cl$^-$.

DOI: https://doi.org/10.7554/eLife.46986.023

The following figure supplement is available for figure 6:

**Figure supplement 1.** Functional properties of mutants of basic residues.

DOI: https://doi.org/10.7554/eLife.46986.024

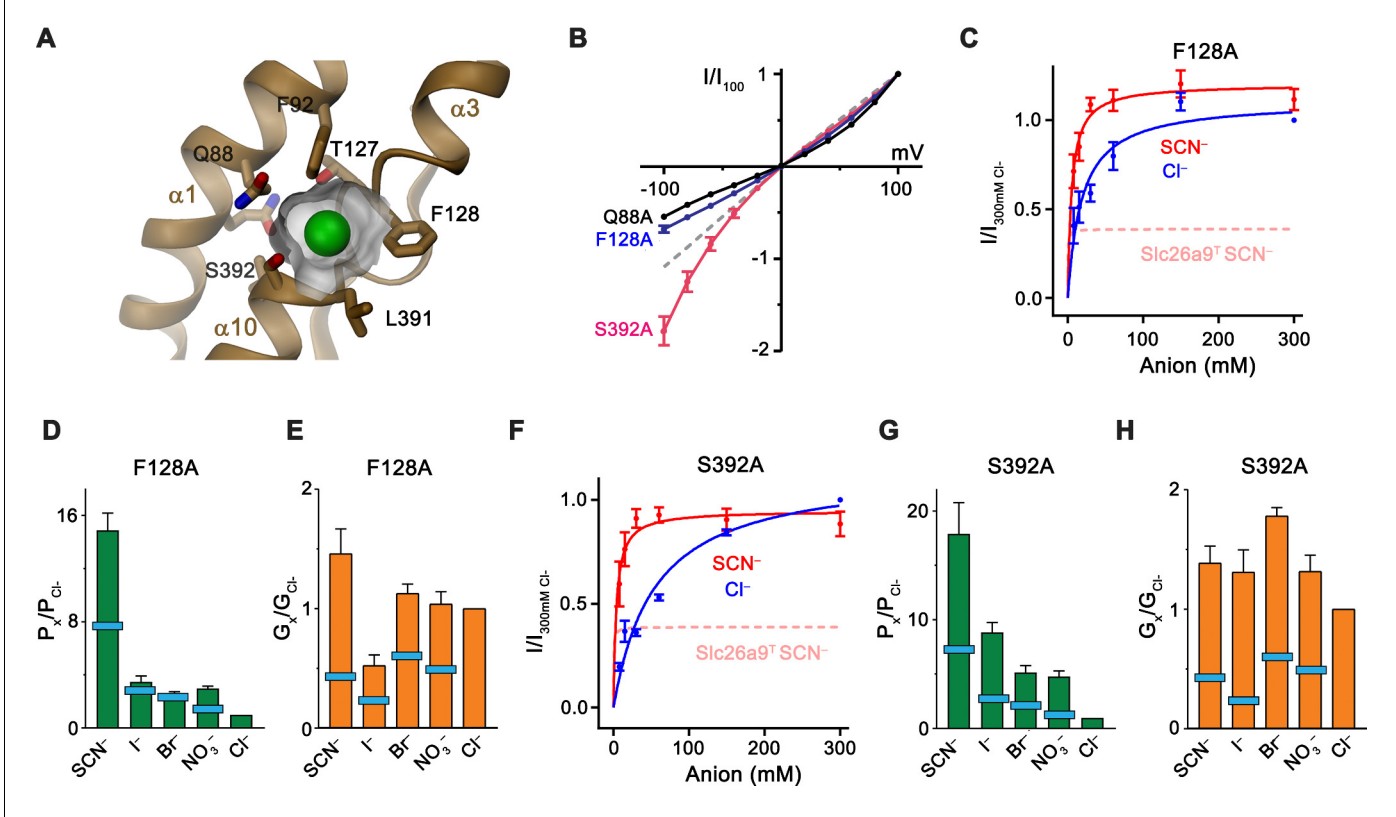

**Figure 7.** Substrate binding site. (**A**) Structure of the Cl⁻ binding site. The molecular surface of the binding pocket is shown. Selected residues are displayed as sticks. Bound Cl⁻ is shown as a green sphere. (**B**) I-V relationships of selected Cl⁻ binding site mutants (Q88A, n = 5; F128A, n = 8; S392A, n = 5). Data for WT Slc26a9$^T$ is shown as dashed line for comparison. (**C**) Conductance-concentration relationships of anion transport across the Slc26a9$^T$ mutant F128A (n = 3). For comparison, SCN⁻ conductance–concentration relationship for non-mutated Slc26a9$^T$ is shown as a pink dotted line. (**D**) Permeability ($P_x/P_{Cl}$) and (**E**) conductance ratios ($G_x/G_{Cl}$) of the Slc26a9$^T$ mutant F128A obtained from bi-ionic substitution experiments (Cl⁻, Br⁻, NO₃⁻, SCN⁻, n = 8; I⁻, n = 3). The light-blue bars indicate corresponding values for WT Slc26a9$^T$. (**F**) Conductance-concentration relationships of anion transport across the Slc26a9$^T$ mutant S392A (n = 5). For comparison, SCN⁻ conductance–concentration relationship for non-mutated Slc26a9$^T$ is shown as a pink dotted line. (**G**) Permeability ($P_x/P_{Cl}$) and (**H**) conductance ratios ($G_x/G_{Cl}$) of the Slc26a9$^T$ mutant S392A obtained from bi-ionic substitution experiments (Cl⁻, Br⁻, n = 5; NO₃⁻, SCN⁻, I⁻, n = 4). The light-blue bars indicate corresponding values for WT Slc26a9$^T$. Data was recorded from excised patches either in symmetric 150 mM Cl⁻ (**B**), the indicated intracellular anion-concentrations with 7.5 mM extracellular Cl⁻ (**C and F**), or at equimolar bi-ionic conditions containing 150 mM extracellular Cl⁻ and 150 mM of the indicated intracellular anion (**D, E, G, H**). Data show mean values of the indicated number of biological replicates, errors are s.e.m.. C-H, Data were recorded at −100 mV.

DOI: https://doi.org/10.7554/eLife.46986.025

The following figure supplements are available for figure 7:

**Figure supplement 1.** I-V-relationships of anion binding site mutants.
DOI: https://doi.org/10.7554/eLife.46986.026
**Figure supplement 2.** Anion selectivity of binding site mutants.
DOI: https://doi.org/10.7554/eLife.46986.027
**Figure supplement 3.** Kinetic properties of anion binding site mutants.
DOI: https://doi.org/10.7554/eLife.46986.028

patch recordings to assess any corresponding changes in transport behavior (*Figure 7—figure supplement 1A–G*). Several mutants altered the linear I-V relationship, with the resulting inward rectification of the mutant S392A indicating a relative increase of the outer barrier and conversely, the outward rectification in the mutants F128A and Q88A, a relative increase of the inner barrier during transport (*Figure 7B*, *Figure 7—figure supplement 1A,D,F*). In contrast, only minor perturbations of the linear I–V relationship are found for the binding site residues F92A and L391A and no change is observed for the mutant N441A residing on the gate module, which, although proximal to the site, is not predicted to participate in ion coordination (*Figure 7—figure supplement 1B,E,G*). The

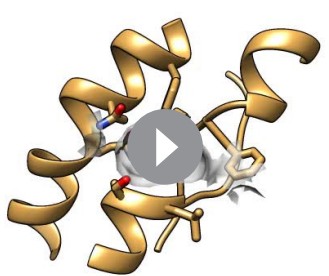

**Video 5.** Structure of the presumed ion binding site. Ribbon representation of the anion binding site with side-chains of interacting residues displayed as sticks. Surface of the binding pocket around a modeled chloride (green sphere) is shown. View is as in *Figure 7A*.

DOI: https://doi.org/10.7554/eLife.46986.030

change in the conductance properties upon mutation of Gln 88 is even more pronounced in the mutant Q88E, which replaces the amino acid conserved in mammalian paralogs with a negatively charged residue found in several prokaryotic transporters (*Geertsma et al., 2015*). Despite the strong apparent surface expression and high stability of the mutant, there are no currents observed in excised patches and severely decreased currents in whole-cell measurements are strongly outward-rectifying, consistent with a compromised transport activity due to the interference of the negatively charged residue with the transported anions (*Figure 7—figure supplement 1H,I*). A striking property of Slc26a9$^T$ is the inverted relationship between permeability and conductivity of transported ions, with larger anions being more permeable but less conductive than Cl$^-$ (*Figure 1D and E*). Although affected in all binding-site mutants (*Figure 7—figure supplements 2* and *3A–E*; *Table 2*), changes are most pronounced for S392A and F128A, where chloride becomes less conductive than other permeant anions, a behavior that is also reflected in the kinetic parameters obtained from the concentration dependence of transport of Cl$^-$ and SCN$^-$ (*Figure 7C–H*; *Figure 7—figure supplement 3*; *Table 2*). A pronounced difference is also observed for the mutant L391A where the $K_m$ of SCN$^-$ increases by nearly two orders of magnitude (*Figure 7—figure supplement 3I*; *Table 2*). Whereas the strong effect of different binding-site mutants on conduction underlines their importance for anion transport, in no case did we detect major increases in permeation by either HCO$_3^-$ or SO$_4^{2-}$, nor did we observe any evidence of coupled transport (*Figure 7—figure supplements 1A–G* and *2*; *Table 2*).

## Discussion

Our study has revealed the architecture of the murine Cl$^-$ transport protein Slc26a9 and it has provided the first direct glimpse at the relationship between structure and function in a mammalian SLC26 family member. Owing to the conservation between paralogs, the observed structural features are likely general for the family. The dimeric organization of Slc26a9 differs markedly from structurally related transport proteins of the SLC4 and SLC23 families, which interact via an extended interface between gate modules (*Alguel et al., 2016*; *Arakawa et al., 2015*; *Chang and Geertsma, 2017*; *Huynh et al., 2018*; *Yu et al., 2017*) (*Figure 8—figure supplement 1*). In contrast, in Slc26a9 the interface between the TMDs is minimal whereas the bulk of subunit interactions are mediated by the cytoplasmic STAS domains involving contacts with the N-terminus and intracellular loop regions of the gate module (*Figure 2*). A similar relationship between transmembrane domains was recently proposed to underlie dimerization of the prokaryotic transporter SLC26Dg in a membrane environment, based on EPR-derived distance constraints (*Chang et al., 2019b*). Although such minimal contacts in the membrane-inserted domain are unusual for oligomeric membrane proteins, they have been recently observed for the mechanosensitive channel OSCA (*Jojoa-Cruz et al., 2018*; *Zhang et al., 2018*), where subunit interactions are mediated exclusively between residues of the cytoplasmic domains.

As previously shown based on whole-cell experiments (*Bertrand et al., 2009*; *Chang et al., 2009b*; *Dorwart et al., 2007*; *Loriol et al., 2008*) and confirmed here with excised patches, Slc26a9 shows large uncoupled macroscopic Cl$^-$ currents that saturate only at high mM concentration. For this reason, the protein has been generally referred to as ion channel. However, such properties would be entirely consistent with a fast bidirectional uniport mechanism as evidenced by other structural and functional data shown in our study. The high $K_m$ of Cl$^-$ matches its intracellular

**Table 2.** Transport properties of WT and mutated $Slc26a9^T$.

| | | WT[*] | Q88A | F92A | T127A | F128A | L391A | S392A | N441A |
|---|---|---|---|---|---|---|---|---|---|
| SCN⁻ | $E_{rev}$ (mV)† | 50.5 | 58.4 | 37.8 | 59.3 | 68.7 | 61.5 | 71.5 | 52.9 |
| | $P_x/P_{Cl}$ | 7.5 | 10.6 | 4.5 | 10.7 | 14.9 | 12.0 | 18.0 | 8.2 |
| | $G_x/G_{Cl}$ | 0.43 | 0.55 | 0.44 | 0.80 | 1.46 | 1.10 | 1.39 | 0.57 |
| I⁻ | $E_{rev}$ (mV) | 26.6 | 16.3 | 14.1 | 9.5 | 31.0 | 23.5 | 53.2 | 29.1 |
| | $P_x/P_{Cl}$ | 2.9 | 1.9 | 1.8 | 1.5 | 3.5 | 2.6 | 8.4 | 3.2 |
| | $G_x/G_{Cl}$ | 0.23 | 0.14 | 0.25 | 0.17 | 0.53 | 0.54 | 1.31 | 0.35 |
| Br⁻ | $E_{rev}$ (mV) | 21.3 | 3.7 | 5.1 | 7.6 | 24.6 | 16.1 | 40.7 | 20.0 |
| | $P_x/P_{Cl}$ | 2.4 | 1.2 | 1.2 | 1.4 | 2.7 | 1.9 | 5.2 | 2.2 |
| | $G_x/G_{Cl}$ | 0.60 | 0.32 | 0.72 | 0.40 | 1.13 | 1.11 | 1.78 | 0.70 |
| $NO_3^-$ | $E_{rev}$ (mV) | 9.4 | 32.6 | 11.2 | 29.0 | 28.5 | 20.7 | 39.1 | 16.9 |
| | $P_x/P_{Cl}$ | 1.5 | 3.7 | 1.6 | 3.2 | 3.1 | 2.3 | 4.8 | 2.0 |
| | $G_x/G_{Cl}$ | 0.49 | 1.20 | 0.52 | 1.46 | 1.04 | 0.70 | 1.31 | 0.77 |
| Cl⁻ | $E_{rev}$ (mV) | −0.1 | −1.4 | −1.5 | −0.5 | 0 | −0.7 | 0.3 | −0.3 |
| | $P_x/P_{Cl}$ | 1 | 1 | 1 | 1 | 1 | 1 | 1 | 1 |
| | $G_x/G_{Cl}$ | 1 | 1 | 1 | 1 | 1 | 1 | 1 | 1 |
| F⁻ | $E_{rev}$ (mV) | −81.1 | −64.2 | −81.0 | −74.5 | −67.6 | −81.0 | −58.8 | −86.7 |
| | $P_x/P_{Cl}$ | .05 | .08 | .04 | .07 | .07 | .05 | 0.11 | 0.03 |
| | $G_x/G_{Cl}$ | nd¶ | nd | nd | nd | nd | nd | nd | nd |
| $HCO_3^-$ | $E_{rev}$ (mV) | −81.7 | −54.4 | −69.5 | −84.7 | −47.3 | −93.1 | −57.4 | −76.0 |
| | $P_x/P_{Cl}$ | 0.05 | 0.12 | 0.07 | 0.04 | 0.16 | 0.03 | 0.10 | 0.05 |
| | $G_x/G_{Cl}$ | nd | nd | nd | nd | nd | nd | nd | nd |
| $SO_4^{2-}$ | $E_{rev}$ (mV) | −96.9 | −102.2 | −78.0 | −104.5 | −70.5 | −93.6 | −71.2 | −82.4 |
| | $P_x/P_{Cl}$ | 0.01 | 0.01 | 0.01 | 0 | 0.02 | 0.01 | 0.02 | 0.01 |
| | $G_x/G_{Cl}$ | nd | nd | nd | nd | nd | nd | nd | nd |
| | $E_{30mM\ Cl}$‡ | −37.7 | −39.3 | −38.5 | −39.8 | −42.1 | −37.9 | −39.0 | −38.3 |
| | $K_{m,Cl}$ (mM) § | 29.5 | 2.3 | 19.2 | 4.7 | 18.6 | 44.0 | 54.1 | 31.3 |
| | $K_{m,SCN}$ (mM) § | 0.5 | 0.2 | 0.8 | 1.6 | 5.2 | 29.4 | 3.8 | 0.7 |
| | $V_{SCN}/V_{Cl}$ § | 0.44 | 0.63 | 0.44 | 0.74 | 1.08 | 1.08 | 0.83 | 0.68 |

[*]WT refers to non-mutated $Slc26a9^T$.

†$E_{rev}$ values, $P_x/P_{Cl}$, and $G_x/G_{Cl}$ were derived from bi-ionic data in **Figure 1—figure supplement 3B**–F, and **Figure 7—figure supplement 2**.

‡Measured reversal potential, in mV, under a 5-fold NaCl gradient ($E_{Cl}$ = –40.2 mV). Extracted from data in **Figure 1C** and **Figure 7—figure supplement 1A–H**.

§$K_m$ and $V_{SCN}/V_{Cl}$ values are derived from conductance–concentration data presented in **Figure 1D**, **Figure 7**, and **Figure 7—figure supplement 3F–J**.

¶nd, not determined.

DOI: https://doi.org/10.7554/eLife.46986.029

concentration and would foster high turnover leading to the observed conduction phenotype. Moreover, transport of the hydrophobic ion SCN⁻ saturates at lower concentration consistent with a mechanism that is mediated by an anion binding site, which would be expected in a transporter. Additionally, our structural data clearly suggest that Slc26a9 works by an alternate-access mechanism akin to other family members. This is evidenced by a comparison between the inward-facing structure of $Slc26a9^T$ determined in detergent, where the ion binding site is accessible to the cytoplasm, and an intermediate conformation of the lipid nanodisc-reconstituted transporter, where an upward movement of the core module has caused the binding site to be buried between core and gate modules (**Figure 8A,B**). In combination, both structures provide insight into conformational changes in which the relatively unconstrained core module, providing the substrate binding site, makes ascending and descending rigid-body movements relative to the immobilized gate module-STAS domain complex, in a process that resembles the elevator transport mechanism (**Drew and Boudker, 2016**; **Ficici et al., 2017**; **Reyes et al., 2009**) (**Figure 8B**). Such mechanism would allow for fast transport kinetics as observed here and as previously described for the Cl⁻/$HCO_3^-$ transporter SLC4A1/Band 3, which shares a similar architecture (**Arakawa et al., 2015**; **Brahm, 1977**). The plausible homology to the known outward-facing structure of SLC4A1/Band 3 (**Arakawa et al., 2015**) also permits the anticipation of the possible extent of conformational changes, where basic residues on the rim of the resulting extracellular vestibule likely create a positive electrostatic

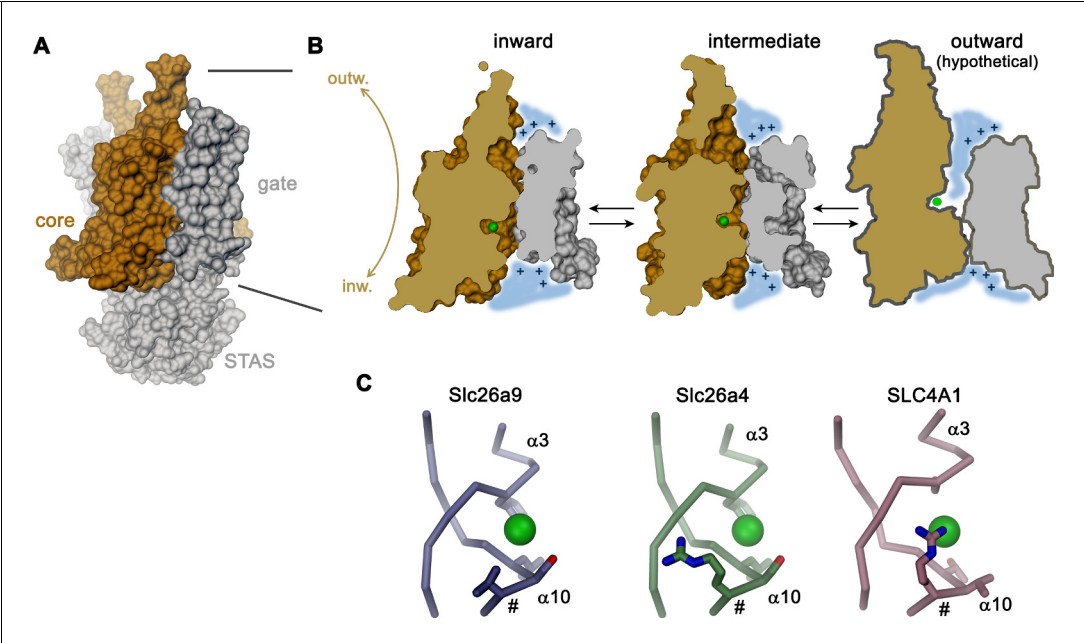

**Figure 8.** Transport mechanism. (**A**) Molecular surface of Slc26a9$^T$ viewed towards the long dimension of the molecule. (**B**) Sections of the TMD in the inward-facing conformation defined by the Slc26a9$^T$ detergent structure (left), an intermediate conformation defined by the Slc26a9$^T$ nanodisc structure (center) and a schematic drawing of a potential outward-facing state (right). (**C**) Putative Cl$^-$-binding site of Slc26a9 (left), a homology model of Slc26a4 (Pendrin, center) and SLC4A1/Band 3 (PDBID: 4YZF). A position that contains a basic residue in SLC26A4 and all mammalian SLC26 paralogs except for Slc26a9 is labeled (#). A and B, STAS domain and gate modules are colored in gray, the core module in beige. A-C, The putative position of a bound Cl$^-$ is indicated as green sphere.

DOI: https://doi.org/10.7554/eLife.46986.031

The following figure supplements are available for figure 8:

**Figure supplement 1.** Dimer architecture of transport proteins sharing the 7 + 7 inverted repeat topology.
DOI: https://doi.org/10.7554/eLife.46986.032

**Figure supplement 2.** Ion binding by SLC26 proteins.
DOI: https://doi.org/10.7554/eLife.46986.033

**Figure supplement 3.** Mechanistic relationships within the SLC26 family.
DOI: https://doi.org/10.7554/eLife.46986.034

environment which is similar to that observed in the vestibule of the inward-facing state. The role of electrostatics in this hypothesis is supported by functional experiments and model-based calculations (*Figure 6A–C*; *Figure 6—figure supplement 1*). During transport, the negatively charged Cl$^-$ is dragged across the transmembrane electric field, thus likely facilitating conformational changes. A similar mechanism may underlie the substrate-conferred electromotility observed in the homologous motor protein Prestin (*Oliver et al., 2001*).

A characteristic feature of Slc26a9 compared to many of its paralogs of the SLC26 family is its distinct anion selectivity and transport mechanism. Unlike several other family members, which are proposed to work as coupled exchangers, transport in Slc26a9 is not coupled to another ion (*Figure 1C*). Moreover, whereas other family members are capable of transporting the divalent anion $SO_4^{2-}$ or the ubiquitous monovalent $HCO_3^-$, neither of these anions is a substrate for electrogenic transport by Slc26a9. The lyotropic permeability sequence of Slc26a9 suggests that ions interact with a low field-strength binding site. The inability to transport $SO_4^{2-}$ is thus probably related to the fact that interactions within the binding site, which does not contain positive countercharges to balance the negatively-charged substrate, may not be sufficiently strong for the coordination of divalent anions. Remarkably, all other mammalian family members harbor very similar predicted binding sites to that of Slc26a9, with the notable exception that they additionally contain a positively charged amino acid at the equivalent position of V393, which is located on the N-terminal end of helix α10

(*Figure 8C*; *Figure 8—figure supplement 2*). Homology models predict this side-chain to reside very close to the substrate binding site (*Figure 8—figure supplement 2*), and an arginine at the equivalent location, which likely contributes to anion interactions, is also found in the $Cl^-/HCO_3^-$ transporter SLC4A1/Band 3 (*Arakawa et al., 2015*) (*Figure 8C*). The inability of Slc26a9 to transport $HCO_3^-$ is particularly remarkable, since the nearly isosteric molecular anion $NO_3^-$ is a transported substrate (*Figure 1E*). Although the $pK_a$ for the deprotonation of $HCO_3^-$ is high (i.e. 10.3) and consequently the concentration of the divalent anion $CO_3^{2-}$ is negligible under physiological conditions, the equilibrium could be perturbed at the binding site, with the large positive electrostatic potential within the aqueous cavity stabilizing the divalent state of the ion, for which interactions with the anion binding site would not be sufficiently strong to make it a transported substrate.

The high homology between mammalian SLC26 paralogs suggests a common general architecture and also implies that other family members may undergo similar conformational transitions to those shown here for Slc26a9$^T$. Therefore, the diversity of functional phenotypes ranging from uncoupled bidirectional uniport as observed in Slc26a9 (and possibly SLC26A7 and/or SLC26A11), to coupled exchangers such as SLC26A2/DTDST and SLC26A4/Pendrin, or even to the motor activity of SLC26A5/Prestin (*Alper and Sharma, 2013*), could rely on comparatively small localized structural differences which in turn impose variable energy barriers for conformational changes (*Figure 8—figure supplement 3*). Uncoupled uniporters like Slc26a9 presumably possess small conformational energy barriers between inward and outward states in the presence or absence of substrate which would permit transition of an unloaded transporter. In contrast, for coupled transport the presumed increase in ion binding affinity paired with an enlarged barrier prohibiting conformational transitions of the unloaded transport domain would itself lead to stoichiometric exchange. In this view, the electromotile motor function of SLC26A5/Prestin could be a consequence of an insurmountable energy barrier preventing the last phase of the transition to an outward-facing state, in line with prior suggestions of SLC26A5/Prestin exhibiting an 'incomplete' transport cycle (*Schaechinger and Oliver, 2007*). While it is currently not possible to pinpoint exact structural determinants of different SLC26 mechanisms, the architectural elements identified here provide a baseline for further studies of SLC26 structure-function relationships.

Since our electrophysiology experiments show instantaneous activity of the transporter that is not obviously dependent on activating ligands (*Figure 1B*), the question concerning the regulation of Slc26a9 activity under physiological conditions is pertinent. This question is particularly important in light of a potential role of human SLC26A9 to compensate for the loss of CFTR mediated $Cl^-$ transport in patients suffering from cystic fibrosis (*Balázs and Mall, 2018*; *Mall and Galietta, 2015*). A potential activation mechanism could be the mobilization of SLC26A9 from intracellular stores, which might be regulated by phosphorylation, as suggested in a previous study (*Dorwart et al., 2007*) and which is also consistent with the predominantly intracellular localization of full-length Slc26a9 observed in our study (*Figure 1—figure supplement 2A*). The fact that removal of the IVS results in significantly greater surface expression suggests that this region may harbor modifiable motif elements which can be involved in surface-trafficking regulation, but this hypothesis must be further evaluated in more physiologically relevant contexts. Another, still unresolved question concerns the precise role of the IVS and the CT in the interaction with accessory proteins including the chloride channel CFTR. The resolution of these questions will be subject of future studies, for which our current work provides a structural and functional foundation.

## Materials and methods

### Key resources table

| Reagent type (species) or resource | Designation | Source or reference | Identifier | Additional information |
|---|---|---|---|---|
| Chemical compound, drug | Pro293S-CDM medium | Lonza | Cat#BE02-025Q | |
| Chemical compound, drug | HyClone HyCell TransFx-H medium | GE Healthcare | Cat#SH30939.02 | |

*Continued on next page*

*Continued*

| Reagent type (species) or resource | Designation | Source or reference | Identifier | Additional information |
|---|---|---|---|---|
| Chemical compound, drug | L-glutamine | Millipore Sigma | Cat#G7513 | |
| Chemical compound, drug | Penicillin-streptomycin | Millipore Sigma | Cat#P0781 | |
| Chemical compound, drug | Fetal bovine serum | Millipore Sigma | Cat#F7524 | |
| Chemical compound, drug | Pluronic F-68 | ThermoFisher Scientific | Cat#24040032 | |
| Chemical compound, drug | Polyethylenimine 25 K MW, linear | Polysciences | Cat#23966–1 | |
| Chemical compound, drug | Dulbecco's Modified Eagle's Medium (DMEM) | Millipore Sigma | Cat#D5546 | |
| Chemical compound, drug | Valproic acid | Millipore Sigma | Cat#P4543 | |
| Chemical compound, drug | cOmplete, EDTA-free Protease Inhibitor Cocktail | Roche | Cat#5056489001 | |
| Chemical compound, drug | Digitonin | AppliChem | Cat#A1905 | |
| Chemical compound, drug | Glyco-diosgenin | Anatrace | Cat#GDN101 | |
| Chemical compound, drug | D-desthiobiotin | Millipore Sigma | Cat#D1411 | |
| Chemical compound, drug | $n$-dodecyl-β-D-maltoside (DDM) | Anatrace | Cat#D310 | |
| Chemical compound, drug | Cholesteryl hemisuccinate, tris salt (CHS) | Anatrace | Cat#CH210 | |
| Chemical compound, drug | 1-palmitoyl-2-oleoyl-sn-glycero-3-phospho ethanolamine (POPE) | Avanti Polar Lipids, Inc | Cat#850757 | |
| Chemical compound, drug | 1-palmitoyl-2-oleoyl-sn-glycero-3-phospho-(1'-rac-glycerol) (POPG) | Avanti Polar Lipids, Inc | Cat#840457 | |
| Chemical compound, drug | Diethyl ether | Millipore Sigma | Cat#296082 | |
| Chemical compound, drug | Triton X-100 | Millipore Sigma | Cat#T9284 | |
| Chemical compound, drug | Biotin | Millipore Sigma | Cat#B4501 | |
| Chemical compound, drug | 9-amino-6-chloro-2-methoxyacridine (ACMA) | Thermo Fisher Scientific | Cat#A1324 | |
| Chemical compound, drug | carbonyl cyanide 3-chlorophenylhydrazone (CCCP) | Merck Millipore | Cat#C2759 | |
| Chemical compound, drug | 4,4'-Diisothiocyanatostilbene-2,2'-disulfonic acid (DIDS) | Millipore Sigma | Cat#D3514 | |
| Commercial assay or kit | QuikChange site-directed mutagenesis kit | Agilent | Cat#200523 | |

*Continued on next page*

*Continued*

| Reagent type (species) or resource | Designation | Source or reference | Identifier | Additional information |
|---|---|---|---|---|
| Commercial assay or kit | NucleoBond Xtra Maxi kit | Macherey-Nagel | Cat#740416 | |
| Commercial assay or kit | StrepTactin Superflow affinity resin slurry | IBA Lifesciences | Cat#2-1206-002 | |
| Commercial assay or kit | Superose 6 10/300 GL | GE Healthcare | Cat#17-5172-01 | |
| Commercial assay or kit | Zorbax GF-450 | Agilent | Cat#884973–902 | |
| Commercial assay or kit | Superose 6 5/150 | GE Healthcare | Cat#29091597 | |
| Commercial assay or kit | Pierce Streptavidin Plus UltraLink Resin | Thermo Fisher Scientific | Cat#53117 | |
| Commercial assay or kit | Bio-Beads SM-2 | Bio-Rad | Cat# 1523920 | |
| Commercial assay or kit | Avestin LiposoFast Liposome Factory Basic | Millipore Sigma | Cat#Z373400 | |
| Commercial assay or kit | 400 nm polycarbonate filters for LiposoFast | Millipore Sigma | Cat#Z373435 | |
| Commercial assay or kit | 96-well black-walled microplate | Thermo Fisher Scientific | Cat#M33089 | |
| Commercial assay or kit | 200 mesh Au 1.2/1.3 cryo-EM grids | Quantifoil | Cat#N1-C14nAu20-01 | |
| Commercial assay or kit | Amicon 100 kDa MWCO centrifugal filter | EMD Millipore | Cat#UFC910008 | |
| Commercial assay or kit | 0.22 µm Ultrafree-MC Centrifugal Filter | EMD Millipore | Cat#UFC30GV | |
| Commercial assay or kit | Borosilicate glass capillary with filament | Sutter Instrument | Cat#BF150-86-10HP | |
| Cell line (human) | HEK293S GnTI- | ATCC | CRL-3022 | |
| Cell line (human) | HEK-293T | ATCC | CRL-1573 | |
| Recombinant DNA | *Mus musculus* Slc26a9 ORF shuttle clone | Source BioScience | ORFeome# OCACo5052B0115D; GenBank BC160193 | |
| Recombinant DNA | pcDNA 3.1 [(+)] vector, Invitrogen | Thermo Fisher Scientific | Cat# V79020 | |
| Recombinant DNA | Modified pcDNA 3.1 vector with C-terminal 3C protease cleavage site, Venus and Myc tags and streptavidin binding peptide | Raimund Dutzler laboratory | N/A | |
| Recombinant DNA | Modified pcDNA 3.1 vector with C-terminal 3C protease cleavage site, Myc tag and streptavidin binding peptide | Raimund Dutzler laboratory | N/A | |
| Recombinant DNA | Expression vector encoding membrane scaffold protein (MSP) E3D1, pMSP1E3D1 | *Denisov et al., 2007* | Addgene, Cat#20066 | |

*Continued on next page*

*Continued*

| Reagent type (species) or resource | Designation | Source or reference | Identifier | Additional information |
|---|---|---|---|---|
| Software, algorithm | SerialEM 3.5.0 | *Mastronarde, 2005* | http://bio3d.colorado.edu/SerialEM/ | |
| Software, algorithm | RELION-3.0 | *Scheres, 2012* | https://www2.mrc-lmb.cam.ac.uk/relion/ | |
| Software, algorithm | CTFFIND4.1 | *Rohou and Grigorieff, 2015* | http://grigoriefflab.jan elia.org/ctf | |
| Software, algorithm | Bsoft 1.9.5 | *Heymann and Belnap, 2007* | https://lsbr.niams. nih.gov/bsoft/ | |
| Software, algorithm | Coot 0.8.8 | *Emsley and Cowtan, 2004* | https://www2.mrc-lmb.cam.ac.uk/person al/pemsley/coot/ | |
| Software, algorithm | PHENIX 1.14 | *Adams et al., 2002* | http://phenix-online.org/ | |
| Software, algorithm | REFMAC5 | *Murshudov et al., 2011* | http://www.ccpem.ac.uk/ | |
| Software, algorithm | MSMS | *Sanner et al., 1996* | http://mgltools.scripps. edu/packages/MSMS/ | |
| Software, algorithm | DINO 0.9.4 | http://www.dino3d.org | http://www.dino3d.org | |
| Software, algorithm | PyMOL 2.3.0 | *DeLano, 2002* | https://pymol.org/2/ | |
| Software, algorithm | Chimera 1.13.1 | *Pettersen et al., 2004* | http://www.cgl.ucsf. edu/chimera/ | |
| Software, algorithm | ChimeraX 0.7 | *Goddard et al., 2018* | https://www.cgl. ucsf.edu/chimerax/ | |
| Software, algorithm | CHARMM | *Brooks et al., 1983* | https://www.charmm. org/charmm/ | |
| Software, algorithm | SWISS-MODEL | *Biasini et al., 2014* | https://swissmodel. expasy.org/ | |
| Software, algorithm | Axon Clampex 10.6 | Molecular Devices | N/A | |
| Software, algorithm | Axon Clampfit 10.6 | Molecular Devices | N/A | |
| Software, algorithm | Prism 7 | GraphPad | https://www.graphpad.com/ | |

## Cell lines

GnTI⁻ cells used for protein expression and purification were obtaiend from ATTC (ATCC CRL-3022). Adherent HEK293T cells used for electrophysiology were obtained from ATTC (ATCC CRL-1573). Both cell-lines were tested negative for mycoplasma contamination. Suspension-adapted HEK293S GnTI⁻ cells expressing murine Slc26a9 were grown at 37°C and 5% CO2 in either Pro293S-CDM or HyClone TransFx-H media, supplemented with 2 mM L-glutamine, 100 U ml$^{-1}$ penicillin/streptomycin, 1% FBS, and 1% Pluronic F-68. Adherent HEK293T cells were grown in DMEM media supplemented with 1 mM L-glutamine, 100 U ml$^{-1}$ penicillin/streptomycin, 10% FBS and 1 mM sodium pyruvate.

## Construct generation

DNA encoding the open reading frame (ORF) for mouse Slc26a9 (GenBank accession: BC160193) was PCR-amplified from a cDNA clone (Source BioScience) and shuttled into a pcDNA 3.1 vector (Invitrogen) which was modified to be compatible with FX cloning technology (*Geertsma and Dutzler, 2011*). Unless stated otherwise, all expression constructs also encoded a C-terminal Rhinovirus 3C protease cleavage site followed by venus YFP (vYFP), a myc epitope tag and a streptavidin binding peptide (SBP), giving the general construct scheme ORF-3C-vYFP-myc-SBP. Assembly of the Slc26a9$^T$ dual-truncation construct entailed removal of the STAS IVS region, via replacement of

residues Pro[558]–Val[660] with a Gly-Ser linker, and deletion of C-terminal residues Pro[745]–Leu[790], akin to a described procedure for the isolated STAS domain from rat Prestin (*Pasqualetto et al., 2010*). Further deletion of N-terminal residues Met[1]–Ala[30] resulted in the construct Slc26a9($\Delta$1-30)[T]. Mutations Q88A, Q88E, F92A, T127A, F128A, L391A, S392A, R205E, K221E, K270E, K431E, and K441E were introduced into Slc26a9[T] using the QuikChange site-directed mutagenesis method (Agilent).

## Protein expression and purification

Suspension-adapted HEK293S GnTI[-] cells (ATCC CRL-3022) were grown in either Pro293S-CDM (Lonza) or HyClone TransFx-H (GE Healthcare) media, supplemented with 2 mM L-glutamine (Sigma), 100 U ml[−1] penicillin/streptomycin (Sigma), 1% FBS, and 1% Pluronic F-68. Cultures were maintained in TubeSpin Bioreactor 600 vessels (TPP), shaken at 185 rpm with an orbital radius of 50 mm, and incubated at 37°C and 5% $CO_2$. For protein expression, a transient transfection protocol relying on 25 kDa linear polyethylenimine (PEI, Polysciences) was employed. One day prior to transfection, cells at high density (3–5 $\times$ 10[6] ml[−1]) were diluted into fresh media to a density of 0.6–0.8 $\times$ 10[6] ml[−1]. Transfection-grade plasmid DNA was purified from MC1061 *E. coli* culture using the NucleoBond Xtra Maxi kit (Macherey-Nagel), and a ratio of 1.3 µg DNA per 10[6] of HEK293S-GnTI[-] cells was used for transfection. Plasmid DNA was diluted into non-supplemented DMEM media (Sigma) at a concentration of 0.015 µg µl[−1], and PEI was added to a concentration of 0.038 µg µl[−1] from a 1 mg ml[−1], pH 7 stock solution. After 10–15 min, the DNA-PEI mixtures were diluted 10-fold directly into the cultured cells, and valproic acid (Sigma) was added to a final concentration of 3 mM. After 40–48 hr, cells were harvested by centrifugation at 500 g for 10 min, washed with PBS, and then either directly used for protein extraction and purification or flash frozen in liquid nitrogen and stored at –80°C.

All subsequent protein extraction and purification procedures were carried out at 4°C. Cell pellets from 4–liter expression batches were resuspended in 60 ml of resuspension buffer (25 mM HEPES, pH 7.4, 200 mM NaCl, 5% glycerol, 2 mM $CaCl_2$, and 2 mM MgCl2, 10 µg ml[−1] DNase, and protease inhibitors (cOmplete EDTA-free, Roche). For cryo-EM analysis of detergent-solubilized Slc26a9[T], the protein was extracted in digitonin (AppliChem), and subsequently purified in the presence of the synthetic digitonin substitute glyco-diosgenin (GDN, Anatrace). To extract membrane proteins, 2% (w/v) digitonin powder was directly dissolved in the cell resuspension, and the mixture was incubated for 1.5 hr under gentle agitation. Insoluble material was removed via ultracentrifugation for 40 min at 150,000 g and the supernatant was passed through a 5 µm syringe filter (Sartorius). The clarified extract was applied to 12 ml of StrepTactin Superflow affinity resin slurry (IBA Lifesciences), which was pre-equilibrated in wash buffer composed of 25 mM HEPES, pH 7.4, 200 mM NaCl, 5% glycerol, and 0.02% GDN. The affinity resin was washed with 20 CV of wash buffer, before elution with 15 ml of wash buffer supplemented with 10 mM D-desthiobiotin (Sigma). The protein was concentrated using a 100 kDa molecular weight cut-off (MWCO) centrifugal filter (Amicon) to 500 µl, typically resulting in a concentration of 1–2 mg ml[−1] fusion protein, and 3C protease was added at a protein: protease mass ratio of 1:2. After a 1 hr incubation, the sample was centrifuged at 10,000 g for 3 min to pellet aggregated material, and the supernatant was passed through a 0.22 µm centrifugal filter (Millipore), before being injected onto a Superose 6 10/300 column (GE Healthcare) equilibrated in SEC buffer, 10 mM HEPES, pH 7.4, 200 mM NaCl, 0.02% GDN. Protein from peak fractions containing cleaved Slc26a9[T] protein was pooled, concentrated to 3 mg ml[−1], and immediately used for cryo-EM grid preparation.

For the preparation of Slc26a9[T] protein to be reconstituted into either liposome or lipid nanodiscs, a similar extraction and purification procedure was performed, with the following modifications. Resuspended cells were extracted with a mixture of 1.5% *n*-dodecyl-β-D-maltoside (DDM, Anatrace) and 0.15% cholesteryl hemisuccinate (CHS, Anatrace), and protein was purified in the presence of 0.03% DDM and 0.003% CHS. For protein which was designated for liposome reconstitution, all other purification procedures were equivalent to the protocol used for cryo-EM sample preparation. However, for nanodisc preparations, an uncleaved variant of Slc26a9[T], lacking a fluorescent fusion protein and therefore consisting of Slc26a9[T]-3C-myc-SBP (henceforth abbreviated as Slc26a9[T]-SBP), was purified and used for nanodisc assembly.

The behavior of all vYFP-tagged Slc26a9 constructs in detergent extracts was assessed with an HPLC system equipped for fluorescence-coupled size exclusion chromatography (FSEC, Agilent), using a Zorbax GF-450 column (Agilent). The same system was also employed to monitor the quality

of purified proteins, using UV detection with a Superose 6 5/150 column (GE Healthcare). All protein samples were purified to $\geq$95% homogeneity, as assessed by SDS-PAGE.

## Liposome and nanodisc reconstitution

For reconstitution of Slc26a9$^T$ into liposomes, a procedure relying on detergent-destabilization of preformed liposomes was employed, as previously described (*Geertsma et al., 2008*). Synthetic POPE and POPG lipids (Avanti Polar Lipids) at a mass ratio of 3:1 POPE:POPG were washed in diethyl ether, dried under $N_2$ in a glass round-bottom flask, and hydrated via gentle sonication in liposome buffer, 10 mM HEPES, pH 7.4, 100 mM KCl, to a concentration of 20 mg ml$^{-1}$. The lipid mixture was subjected to three freeze-thaw cycles, followed by extrusion through two 400 nm poly-carbonate filters (LiposoFast Basic, Avestin) to form large unilamellar vesicles (LUVs). LUVs were diluted in liposome buffer to 4 mg ml$^{-1}$, and 10% (w/v) Triton X-100 was added dropwise until the solution absorbance at 540 nm value reached a maximum, indicating suitable destabilization of liposomes for incorporation of membrane protein. Purified Slc26a9$^T$ (in DDM-CHS) was added to the destabilized LUVs at a protein:lipid ratio of 1:80 (w/w). After 20 min of incubation at RT, the sample was cooled to 4°C and detergent was removed via sequential additions of 250 mg SM-2 Bio-Beads (Bio-Rad) per 5 ml, every 24 hr for three days. Bio-Beads were removed via gravity filtration, and Slc26a9$^T$ proteoliposomes were pelleted via ultracentrifugation at 150,000 g for 30 min, resuspended in liposome buffer to 20 mg ml$^{-1}$, frozen in liquid $N_2$, and stored at $^-$80°C. Identical procedures were used to produce mock liposomes lacking any reconstituted membrane protein.

Slc26a9$^T$-SBP was reconstituted into lipid nanodiscs as described (*Ritchie et al., 2009*), with subtle modifications. We utilized the engineered membrane scaffold protein (MSP) MSP1-E3D1 because the estimated resultant nanodisc dimeter of 12 nm is ideal for the incorporation of Slc26a9$^T$, which our preliminary cryo-EM analysis had suggested to possess a length of 10–11 nm when solubilized in detergent. MSP1-E3D1 was purified as described (*Ritchie et al., 2009*) and a 3:1 (w/w) mixture of the synthetic lipids POPC:POPG was dried and hydrated in 10 mM HEPES, pH 7.4, 100 mM KCl, as described above for proteoliposome preparations, except the final lipid concentration was 10 mM, and DDM was added to the stock lipid mixture to a final concentration of 27.5 mM. To assemble nanodiscs, purified Slc26a9$^T$-SBP was diluted into 10 mM HEPES, 200 mM NaCl, 0.5 mM DDM, to a final protein concentration of 6 µM. Lipids were added to a concentration of 1.9 mM, and the mixture was incubated on ice for 30 min. Next, MSP1-E3D1 was added to a concentration of 20 µM, giving a final protein:lipid:MSP molar ratio of 1:450:20, at a final volume of 750 µl. After an additional 30 min of incubation on ice, 200 mg of SM2 Bio-Beads was added to remove detergent, and the reaction was allowed to incubate overnight under slow rotation at 4°C. Bio-beads were removed via gravity filtration, and Slc26a9$^T$-SBP-reconstituted nanodiscs were isolated from empty nanodiscs via secondary purification with 3 ml of Streptavidin Plus UltraLink affinity resin (ThermoFisher) using detergent-free nanodisc buffer, 10 mM HEPES, pH 7.4, 150 mM NaCl, and elution with 5 mM biotin. Finally, nanodiscs were injected onto a Superose 6 5/300 column equilibrated in nanodisc buffer, and a peak containing Slc26a9$^T$-SBP nanodiscs was concentrated to 1 mg ml$^{-1}$ and immediately used for cryo-EM grid preparation.

## Liposome anion transport assay

To measure electrogenic anion transport in Slc26a9$^T$ proteoliposomes, we used a method based on a previously described fluorometric assay (*Kane Dickson et al., 2014*) in which the internal liposome buffer ideally contains no permeant ions. Since Slc26a9 has low permeability to cations and sulfate, we exchanged the internal buffer of Slc26a9$^T$ proteoliposomes and mock liposomes to 10 mM HEPES, 50 mM $Na_2SO_4$ by pelleting the proteoliposomes (150,000 g, 30 min), resuspending in internal sulfate buffer to a concentration of 1 mg ml$^{-1}$, and subjecting the sample to three freeze-thaw cycles. Finally, liposomes were again pelleted and resuspended to 20 mg ml$^{-1}$ in internal sulfate buffer. All subsequent procedures were carried out at RT to restrict formation of multilamellar vesicles, and mock liposomes were always assayed in parallel as a negative control. To form small unilamellar vesicles, 20 µl aliquots of liposomes in 0.2 ml conical tubes were placed in a bath sonicator for 5–10 s, until the opaque solution became translucent. Liposomes were diluted 100-fold into flux buffer, consisting of 10 mM HEPES, pH 7.4, 75 mM NaCl, and 2 µM of the fluorophore 9-amino-6-chloro-2-methoxyacridine (ACMA, ThermoFisher), and 100 µl aliquots were transferred to a 96-

well black-walled microplate (ThermoFisher). ACMA fluorescence was monitored with an Infinite M1000 spectrofluorometer (Tecan) in 5 s intervals, using excitation and emission wavelengths of 412 nm and 482 nm, respectively. After recording baseline fluorescence for 60 s, data collection was paused, 300 nM of the proton ionophore carbonyl cyanide 3-chlorophenylhydrazone (CCCP, Sigma) was added, and fluorescence measurements were immediately continued. Fluorescence intensity for each experimental condition was normalized to the initial value directly following addition of CCCP. For inhibition experiments, diluted proteoliposomes (0.2 mg ml$^{-1}$) were pre-incubated for 5 min with 0–250 µM 4,4′−2,2′-disulfonic acid (DIDS, Sigma) and dimethyl sulfoxide (DMSO) was added to all samples to a final concentration of 1% to maintain solubility of DIDS.

## Cryo-EM sample preparation and data collection

For structure determination of Slc26a9$^T$ by cryo-EM, 2.5 µl samples of GDN-purified protein at a concentration of 3 mg ml$^{-1}$ were applied to glow-discharged holey carbon grids (Quantifoil R1.2/1.3 Au 200 mesh). For the structural characterization of the protein in a membrane-like environment, 2.5 µl samples of Slc26a9$^T$ reconstituted in E3D1 lipid nanodiscs at a concentration of 0.5–1 mg ml$^{-1}$ were applied in a similar manner. Excess liquid was removed in a controlled environment (4°C and 100% relative humidity) by blotting grids for 3–6 s. Grids were subsequently flash frozen in liquid propane-ethane mix using a Vitrobot Mark IV (Thermo Fisher Scientific). Slc26a9$^T$ in detergent (dataset 1) was imaged in a 300 kV Titan Krios (Thermo Fisher Scientific) with a 100 µm objective aperture. Slc26a9$^T$ in nanodiscs (dataset 2) was recorded on a 300 kV Tecnai G$^2$ Polara (FEI) with a 100 µm objective aperture. All data were collected using a post-column quantum energy filter (Gatan) with a 20 eV slit and a K2 Summit direct detector (Gatan) operating in super-resolution (for dataset 1) and counting (for dataset 2) modes. Dose-fractionated micrographs were recorded in an automated manner using SerialEM (*Mastronarde, 2005*) with a defocus range of –0.5 to –3.0 µm. Dataset one was recorded at a nominal magnification of 46,511 corresponding to a pixel size of 1.075 Å/pixel (0.5375 Å/pixel in super-resolution) with a total exposure time of 13 s (65 individual frames) and a dose of approximately 1.1 e$^-$/Å$^2$/frame. Dataset two was recorded at a nominal magnification of 37,313 corresponding to a pixel size of 1.34 Å/pixel with a total exposure time of 12.5 s (50 individual frames) and a dose of approximately 1.2 e$^-$/Å$^2$/frame. The total electron dose on the specimen level for dataset 1 and 2 was approximately 70 e$^-$/Å$^2$ and 60 e$^-$/Å$^2$, respectively.

## Cryo-EM image processing

The recorded super-resolution images of Slc26a9$^T$ in detergent (dataset 1) were down-sampled twice by Fourier cropping and all individual frames were used for correction of beam-induced movement using a dose-weighting scheme in RELION's own implementation of MotionCor2 algorithm available in version 3.0 (*Zivanov et al., 2018a*). The CTF parameters were estimated on summed movie frames using CTFFIND4.1 (*Rohou and Grigorieff, 2015*). All individual frames of Slc26a9$^T$ reconstituted in nanodiscs (dataset 2) were pre-processed in the same manner. Low-quality micrographs showing a significant drift or poor CTF estimates were discarded resulting in datasets of 2838 images of Slc26a9$^T$ in detergent and 3134 images of Slc26a9$^T$ in nanodiscs, which were subjected to further data processing in RELION (*Scheres, 2012*). From dataset one 416,164 particles were picked automatically using low-pass filtered 2D templates generated from an initial reference-free 2D classification. The particles were extracted with a box size of 232 pixels, down-scaled twice and subjected to a couple of rounds of 2D classification. Having discarded false positives and particles of poor quality, the dataset was reduced to 241,438 particles. The initial 3D reconstruction, which was generated from 3600 randomly chosen cleaned particles, was low-pass filtered to 60 Å and used as a template in a subsequent 3D classification. Multiple independent rounds of non-symmetrized 3D classification with a different number of classes were performed in order to isolate the most homogeneous subset of particles. One out of five classes, contained almost two-thirds of all the particles and showed clear two-fold symmetry. 157,644 particles belonging to this class were subjected to auto-refinement with imposed C2 symmetry. The particles were then unbinned to a pixel size of 1.075 Å/pixel and refined to 4.3 Å using C2 symmetry and a soft mask around the protein-detergent micelle density. The reconstruction was further improved to 4.09 Å by performing per-particle defocus and beam tilt corrections followed by Bayesian polishing (*Zivanov et al., 2018a*; *Zivanov et al., 2018b*). To enhance features of the peripheral helices α6 and α7, which were

of lower local resolution than the core of the transmembrane domain, the density representing the detergent belt was subtracted from individual particles. In silico modified particles were used as an input in a subsequent masked 3D classification (*Scheres, 2016*). To separate any remaining heterogeneity particle angles were kept the same as the orientations from the consensus model. The final class containing 112,930 homogeneous particles was further auto-refined with C2 symmetry and in the presence of a soft mask around the protein density. The final map at 3.96 Å was sharpened using an isotropic b-factor of −205 Å$^2$. The initial dataset of Slc26a9$^T$ in E3D1 lipid nanodiscs contained 711,032 particles, which were subjected to 2D classification in order to discard particles of poor quality, small aggregates and any remaining empty nanodiscs. After a stringent selection 308,872 particles were used in a subsequent non-symmetrized 3D classification. Similarly to the dataset of Slc26a9$^T$ in detergent, multiple independent runs were performed to optimally separate the heterogeneity in the dataset. Here, 3D classes showed more apparent heterogeneity compared to the detergent dataset. One out of seven classes containing 64,670 particles displayed well-resolved two-fold symmetric STAS domains although the extracellular loops located between the helices α3 and α4 of each subunit showed poorer density indicating increased flexibility of this region. As at this level of 3D classification the quality of the reconstruction did not allow to observe whether this flexibility is further extended to the TM domain or whether is affecting one or two monomers, further refinement was continued, side-by-side, without imposing any symmetry as well as with imposing C2 symmetry. Within the auto-refined non-symmetrized class one monomer was better resolved comparing to the second monomer and it also showed similar arrangement as either monomer in the structure of the protein in detergent. On the other hand, the second monomer showed increased flexibility only in the TM domain and the extracellular region as its STAS domain was identical as in the first monomer and as the STAS domain in the detergent structure. These results have suggested that despite eliminating particles with large heterogeneous differences during previous rounds of 2D and 3D classification, small conformational changes accounted for the remaining flexibility. In order to separate these changes into discrete classes a focused 3D classification was performed (*Scheres, 2016*). In this approach the mask around the flexible monomer was applied and the angles were kept fixed at the orientations from the auto-refined model. Given the classification focused only on a small region of the protein, the regularization parameter T was increased to 40. Approximately one-fourth of the remaining particles from both non-symmetrized and C2-symmetrized consensus models contributed to a class that recovered the density representing the extracellular loop. After identification of these homogeneous subsets that represent the stable monomer, final 3D auto-refinement of the whole dimeric particles was performed either with C1 or C2 symmetry imposed, yielding in both cases two-fold symmetric reconstructions of 8.4 Å and 7.77 Å, respectively. In all cases resolution was estimated in the presence of a soft solvent mask and based on the gold standard Fourier Shell Correlation (FSC) 0.143 criterion (*Chen et al., 2013*; *Rosenthal and Henderson, 2003*; *Scheres, 2012*; *Scheres and Chen, 2012*). High-resolution noise substitution was applied to correct FSC curves for the effect of soft masking in real space (*Chen et al., 2013*). The local resolution was estimated using BlocRes from the Bsoft package (*Cardone et al., 2013*; *Heymann and Belnap, 2007*).

## Model building and refinement

The model of Slc26a9$^T$ in the inward-facing state was built in Coot (*Emsley and Cowtan, 2004*) using the transmembrane domain structure of SLC26Dg (PDBID: 5DA0) and the chicken Prestin STAS domain (PDBID: 5EZB) as templates. The cryo-EM density of Slc26a9$^T$ in detergent was of sufficiently high resolution to unambiguously assign residues 5–27, 42–559 and 661–740. The model was improved iteratively by cycles of real-space refinement in PHENIX (*Adams et al., 2002*) with secondary structure and 2-fold NCS constraints applied, reciprocal-space refinement in REFMAC5 (*Brown et al., 2015*; *Murshudov et al., 2011*) and manual corrections in Coot. Validation of the refinement of the model was performed in REFMAC5, distributed as part of the CCP-EM suite (*Burnley et al., 2017*), and represented as Fourier Shell Correlation (FSC$_{sum}$) between the refined model and the corresponding final cryo-EM density map. For cross-validation the detergent cryo-EM dataset was split into two subsets which were used to calculate two independent maps (half map one and half map 2). To detect possible overfitting and hence over-estimation of the resolution, random shifts, up to 0.5 Å, were applied to the coordinates of the model of the inward-facing state, which was then refined against the unfiltered half map 1. The cross-validation was done by

comparing the $FSC_{work}$ (estimated for the shaken-refined model and half map 1) and the $FSC_{free}$ (estimated for the shaken-refined model and half map 2, which was not used in the refinement). The model of the intermediate state of $Slc26a9^T$ in nanodiscs was assembled by splitting the model of the inward-facing state into six independent entities, the STAS, gate and core domains, and refining them in PHENIX as rigid-bodies. In the next stage, the STAS, gate and core domain were linked into a single polypeptide chain and refined as described above with global minimization applied. The secondary structure and NCS constrains were maintained at all times. As the protein side-chains were not defined in our 7.77 Å cryo-EM reconstruction of the intermediate state, we have truncated all the side-chains to alanine. The validation of the refinement was performed as described above and represented as $FSC_{sum}$. Owing to the low-resolution of the 3D reconstruction from the dataset in nanodiscs, $FSC_{work}$ and $FSC_{free}$ were not calculated. Surfaces were calculated with MSMS (*Sanner et al., 1996*). Figures and videos containing molecular structures and densities were prepared with DINO (http://www.dino3d.org), PyMOL (*DeLano, 2002*), Coot (*Emsley and Cowtan, 2004*),Chimera (*Pettersen et al., 2004*) and ChimeraX (*Goddard et al., 2018*).

## Modeling and Poisson-Boltzmann calculations

The electrostatic potential in the intracellular vestibule leading to the $Cl^-$-binding site was calculated by solving the linearized Poisson–Boltzmann equation in CHARMM (*Brooks et al., 1983*; *Im et al., 1998*) on a 150 Å $\times$ 170 Å $\times$ 200 Å grid (1 Å grid spacing) followed by focusing on a 120 Å x 130 Å x 160 Å grid (0.5 Å grid spacing). Partial protein charges were derived from the CHARMM36 all-hydrogen atom force field. Hydrogen positions were generated in CHARMM. Histidines were protonated. The protein was assigned a dielectric constant ($\epsilon$) of 2. Its transmembrane region was embedded in a 30 Å-thick slab ($\epsilon = 2$) representing the hydrophobic core of the membrane and two adjacent 15 Å-thick regions ($\epsilon = 30$) representing the headgroups. The membrane region contained a 22 Å-high and 30 Å-wide aqueous cylinder ($\epsilon = 80$) covering the intracellular vestibule of the protein and was surrounded by an aqueous environment ($\epsilon = 80$). Calculations were carried out in either 150 mM of monovalent mobile ions in the aqueous regions (except for the membrane-inserted cylinder). The electrostatic surface potential shown in *Figure 3—figure supplement 2F* was calculated and displayed with COOT (*Emsley and Cowtan, 2004*). Homology models of murine SLC26 paralogs were prepared with the SWISS-MODEL homology modelling server (*Biasini et al., 2014*).

## Electrophysiology

Adherent HEK293T cells (ATCC CRL-1573) were grown in DMEM media supplemented with 1 mM L-glutamine (Sigma), 100 U ml$^{-1}$ penicillin/streptomycin (Sigma), 1 mM sodium pyruvate (Sigma), and 10% FBS, at 37°C and 5% $CO_2$, in 6 cm culture dishes. For transfection, 2.5–5 µg plasmid DNA encoding the construct of interest was mixed with 25 kDa linear polyethylenimine (Polysciences) in a ratio of 1:2.5 (w/w) in 0.3 ml PBS, incubated for 10 min at room temperature, and added dropwise to cells. Transfected cells were used within 24 hr for whole-cell recordings, or within 40 hr for excised patch recordings. Borosilicate glass capillaries (OD = 1.5 mm, ID = 0.86 mm, Sutter) were pulled and fire-polished, giving patch pipettes with resistances of 3–8 MΩ when backfilled with 150 mM NaCl pipette solution. All voltage-clamp signals were recorded using an Axopatch 200B amplifier (Molecular Devices) digitized with a Digidata 1440A A/D converter (Molecular Devices), filtered at 5 kHz, sampled at 20 kHz, and collected with Clampex 10.6 (Molecular Devices). For full-length Slc26a9, membrane seal resistance was typically 2–10 GΩ. However, for $Slc26a9^T$, membrane seal resistance was frequently <1 GΩ, owing to its considerably higher constitutive $Cl^-$ conductance. Membrane seals for $Slc26a9^T$-expressing cells showing resistance <0.2 GΩ were rejected. In whole-cell measurements of full-length Slc26a9, typical cell capacitance upon break-in was 15–30 pF, and series resistance was <10 MΩ after 60–80% compensation. Capacitive transients and series resistance could not be measured for $Slc26a9^T$ upon break-in, presumably due to its high constitutive conductance. Therefore, $Slc26a9^T$ whole-cell currents could not be expressed as normalized current densities (pA pF$^{-1}$) and instead average raw currents were used for comparisons. For all experiments, the typical data acquisition protocol consisted of a holding potential of 0 mV for 0.2 s, followed by 0.2–0.4 s voltage steps from –100 mV to +100 mV in 20 mV increments, before returning to 0 mV. Calculated liquid junction potentials (JPCalcW, Molecular Devices) were corrected if they were greater than 2 mV, but never exceeded 5 mV. For all experiments, cells or excised patches expressing Slc26a9

constructs were only accepted for analysis if they displayed negligible conductance of NMDG-methylsulfonate.

Excised inside-out patches were employed for all ion selectivity experiments, in which the standard pipette solution contained 146 mM NaCl, 2 mM MgCl$_2$, and 5 mM EGTA 10 mM HEPES, pH 7.4. For all solutions, pH was adjusted using NMDG or methanesulfonic acid. For assessment of anion/cation selectivity, patches were intracellularly perfused with solutions containing the same components as the pipette solution, except the NaCl concentration was varied between 15–150 and balanced to the ion concentrations of the pipette solution with NMDG-methylsulfonate. When employing perfusate concentrations above 150 mM NaCl, KCl was substituted for in order to minimize the liquid junction potential. For bi-ionic anion substitution experiments, the intracellular solution contained 150 mM sodium salts of substitute anions Br$^-$, I$^-$, F$^-$, SCN$^-$, NO$_3^-$, and SO$_4^{2-}$, as well as 2 mM Mg(OAc)$_2$, 5 mM EGTA, and 10 mM HEPES, pH 7.4. For the measurement of bicarbonate selectivity, a perfusate concentration of 165 mM NaHCO$_3$ was prepared, because [HCO$_3^-$] / [HCO$_3^-$] + [H$_2$CO$_3$ + CO$_2$]=0.9 at pH 7.4, due to the pK$_a$ of 6.4 for protonation of HCO$_3^-$. Fresh NaHCO$_3$ solution was prepared every 1–3 hr, and pH monitoring of HCO$_3^-$ solutions indicated that the decrease in [HCO$_3^-$] due to production of CO$_2$ gas never exceeded 20% during recordings. In all selectivity experiments, P$_x$/P$_{Cl}$ values were calculated according to the GHK equation, and I-V plots were normalized to the current at +100 mV in symmetrical 150 mM NaCl. To obtain G$_x$/G$_{Cl}$ conductance ratios, macroscopic chord conductance was calculated according to the equation $G = \frac{I}{(V_m - E_{rev})}$, using the efflux current at –100 mV. For the determination of K$_m$ and relative V$_{max}$ values for Cl$^-$ and SCN$^-$, the pipette solution contained 7.5 mM NaCl, 142.5 mM NMDG-methylsulfonate, 2 mM Mg(OAc)$_2$, 5 mM EGTA, and 10 mM HEPES, pH 7.4, and the perfusion solutions contained 7.5–300 mM chloride or thiocyanate salts (sodium salts were used for all solutions except for 300 mM KCl). The current amplitude at 300 mM intracellular Cl$^-$ and –100 mV was used to normalize datasets. To calculate macroscopic SCN$^-$ conductances, E$_{rev}$ values for each perfusion concentration of SCN$^-$ were calculated according to the GHK equation, using P$_{SCN}$/P$_{Cl}$ values which were earlier determined in bi-ionic 150 mM SCN$_{in}$/Cl$_{out}$ selectivity experiments. Fitting the conductance vs. anion concentration data with the Michaelis-Menten equation allowed determination of V$_{max}$ and K$_m$ values. All electrophysiology data were analyzed with Clampfit 10.6 (Molecular Devices), Excel (Microsoft), and Prism 7 (GraphPad).

## Statistics and reproducibility

Electrophysiology data were repeated multiple times from different transfections with very similar results. Conclusions of experiments were not changed upon inclusion of further data. In all cases, leaky patches were discarded.

## Accession codes

The cryo-EM density maps of Slc26a9$^T$ in detergent and lipid nanodiscs have been deposited in the Electron Microscopy Data Bank under ID codes EMD-4997 and EMD-4997, respectively. The coordinates for the atomic model of Slc26a9$^T$ refined against the 3.96 Å cryo-EM density in detergent and the coordinates of Slc26a9$^T$ with side-chains truncated to their β-positions refined against the 7.77 Å cryo-EM density in lipid nanodiscs have been deposited in the Protein Data Bank under ID codes 6RTC and 6RTF, respectively.

## Acknowledgements

We thank O Medalia and M Eibauer, the Center for Microscopy and Image Analysis (ZMB) of the University of Zurich, and the Mäxi foundation for the support and access to the electron microscopes, S Klauser and S Rast for their help in establishing the computer infrastructure. All members of the Dutzler lab are acknowledged for their help at various stages of the project. This research was supported by a grant from the Swiss National Science Foundation (No. 31003A_163421).

## Additional information

### Funding

| Funder | Grant reference number | Author |
|---|---|---|
| Schweizerischer Nationalfonds zur Förderung der Wissenschaftlichen Forschung | 31003A_163421 | Raimund Dutzler |

The funders had no role in study design, data collection and interpretation, or the decision to submit the work for publication.

### Author contributions

Justin D Walter, Conceptualization, Data curation, Formal analysis, Validation, Investigation, Visualization, Methodology, Writing—original draft, Writing—review and editing, Generated expression constructs, purified and reconstituted protein, Recorded and analyzed electrophysiology and proteoliposome transport data; Marta Sawicka, Conceptualization, Data curation, Formal analysis, Validation, Investigation, Visualization, Methodology, Writing—original draft, Writing—review and editing, Prepared the samples for cryo-EM, collected cryo-EM data and proceeded with structure determination; Raimund Dutzler, Conceptualization, Formal analysis, Supervision, Funding acquisition, Visualization, Writing—original draft, Project administration, Writing—review and editing

### Author ORCIDs

Justin D Walter (iD) https://orcid.org/0000-0002-1492-3055
Marta Sawicka (iD) https://orcid.org/0000-0003-4589-4290
Raimund Dutzler (iD) https://orcid.org/0000-0002-2193-6129

### Decision letter and Author response

Decision letter https://doi.org/10.7554/eLife.46986.051
Author response https://doi.org/10.7554/eLife.46986.052

## Additional files

### Supplementary files

• Transparent reporting form
DOI: https://doi.org/10.7554/eLife.46986.035

### Data availability

The cryo-EM density maps of Slc26a9T in detergent and lipid nanodiscs have been deposited in the Electron Microscopy Data Bank under ID codes EMD-4997 and EMD-4997, respectively. The coordinates for the atomic model of Slc26a9T refined against the 3.96 Å cryo-EM density in detergent and the coordinates of Slc26a9T with sidechains-truncated to their β-positions refined against the 7.77 Å cryo-EM density in lipid nanodiscs have been deposited in the Protein Data Bank under ID codes 6RTC and 6RTF.

The following datasets were generated:

| Author(s) | Year | Dataset title | Dataset URL | Database and Identifier |
|---|---|---|---|---|
| Walter JD, Sawicka M, Dutzler R | 2019 | Cryo-EM density of murine Solute Carrier 26 family member A9 (Slc26a9) anion transporter in the inward-facing state | https://www.ebi.ac.uk/pdbe/entry/emdb/EMD-4997 | Electron Microscopy Data Bank, EMD-4997 |
| Walter JD, Sawicka M, Dutzler R | 2019 | Cryo-EM density of murine Solute Carrier 26 family member A9 (Slc26a9) anion transporter in an intermediate state | https://www.ebi.ac.uk/pdbe/entry/emdb/EMD-4998 | Electron Microscopy Data Bank, EMD-4998 |
| Walter JD, Sawicka M, Dutzler R | 2019 | Structure of murine Solute Carrier 26 family member A9 (Slc26a9) | https://www.rcsb.org/structure/6RTC | Protein Data Bank, 6RTC |

| | | anion transporter in the inward-facing state | | |
|---|---|---|---|---|
| Walter JD, Sawicka M, Dutzler R | 2019 | Structure of murine Solute Carrier 26 family member A9 (Slc26a9) anion transporter in an intermediate state | https://www.rcsb.org/structure/6RTF | Protein Data Bank, 6RTF |

The following previously published datasets were used:

| Author(s) | Year | Dataset title | Dataset URL | Database and Identifier |
|---|---|---|---|---|
| Geertsma ER, Chang YN, Shaik FR, Neldner Y, Pardon E, Steyaert J, Dutzler R | 2015 | Structure of the the SLC26 transporter SLC26Dg in complex with a nanobody | http://www.rcsb.org/structure/5DA0 | Protein Data Bank, 5DA0 |
| Alguel Y, Arakawa T, Yugiri TK, Iwanari H, Hatae H, Iwata M, Abe Y, Hino T, Suno CI, Kuma H, Kang D, Murata T, Hamakubo T, Cameron AD, Kobayashi T, Hamasaki N, Iwata S | 2015 | Crystal structure of the anion exchanger domain of human erythrocyte Band 3 | http://www.rcsb.org/structure/4YZF | Protein Data Bank, 4YZF |
| Alguel Y, Amillis S, Leung J, Lambrinidis G, Capaldi S, Scull NJ, Craven G, Iwata S, Armstrong A, Mikros E, Diallinas G, Cameron AD, Byrne B | 2016 | The structure of the eukaryotic purine/H+ symporter, UapA, in complex with Xanthine | https://www.rcsb.org/structure/5I6C | Protein Data Bank, 5I6C |

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
