## [Decision Letter]

Thank you for submitting your article "Cryo-EM structures of Slc26a9 reveal mechanism of uncoupled chloride transport" for consideration by *eLife*. Your article has been reviewed by three peer reviewers, and the evaluation has been overseen by a Reviewing Editor and Richard Aldrich as the Senior Editor. The following individuals involved in review of your submission have agreed to reveal their identity: Alexandria N Miller (Reviewer #2); Seok-Yong Lee (Reviewer #3).

The reviewers have discussed the reviews with one another and the Reviewing Editor has drafted this decision to help you prepare a revised submission.

Summary:

The authors report the structural and functional analysis of a mammalian anion transporter SLC26A9. Slc26a9 is an epithelial anion transporter/channel that is highly expressed in lung and stomach epithelia with reported roles in airway clearance and gastric acid production. In humans, mutations to the gene encoding Slc26a9 have been associated with increased severity to airway clearance diseases such as cystic fibrosis. The cryo-EM structures of the mouse Slc26a9^T^ were solved in detergent (GDN) to ~4Å resolution, and in a nanodisc preparation to ~8Å resolution. This is the first structure of a eukaryotic SLC26 transporter and follows this group's previous work on the bacterial SLC26 transporter. Both structures are dimeric, and the majority of the dimeric interface is located within a conserved cytosolic STAT domain, a potential interaction platform with accessory proteins such as CFTR. Comparison of GDN-purified and nanodisc-reconstituted Slc26a9^T^ structures suggest that they represent inward-facing (GDN) and inward-occluded (nanodisc) states of a transporter. Electrophysiological analysis in combination with the structure shows that the SLC26A9 mediates uncoupled Cl^-^ transport. In contrast to other studies in the literature, the authors demonstrate that these channels have a strong preference for chloride as opposed to bicarbonate. Structure-guided mutagenesis identifies the residues lining the ion transport pathway. Overall, this is an excellent study combining cryo-EM analysis with rigorous functional studies of a eukaryotic SLC26 family transporter/channel. All the reviewers agree this is a very important study but have raised some concerns mainly with regards to the interpretation and presentation of the data.

Essential revisions:

1) Much of the discussion between the reviewers focused on whether there is enough evidence to establish one way or the other that SLC26A9 is a channel or a transporter. Although your structural data is compatible with a transporter model, the difference between a channel and transporter comes down to the transport rate. According to the classical definition, ion transport through a channel is diffusion limited whereas a transporter is about two to three orders of magnitude slower. Therefore, this question can only be addressed by measuring unitary fluxes. Instead, the reviewers are suggesting that the Discussion section should be reframed to provide a more balanced view of the mode of ion transport.

2) The fit of the cryo-EM density to the model: I suggest the authors show the cryo-EM density for the core domain and gate domains, and show how well the structures of the transporter in the inward-open and (putative) inward-occluded states fit the density. This way readers will appreciate that the rigid body movement of the core domain does indeed occur between the two states. It is obvious to me that the rigid body motion could explain the conformational differences between the two models when I inspected the provided maps and the coordinates, which was not obvious by just reading the manuscript.

Along the same line, authors need to provide more figures to show why the nanodisc reconstruction is considered as inward-occluded. Authors provided Figure 5D for comparison, but that figure just shows that there is a rigid body motion between the structures. I suggest showing surface representation like Figure 7A and B in Figure 5 or supplementary figure.

3) Due to the low resolution of the nanodisc-reconstituted reconstruction, I would be cautious to qualify the state as “inward-occluded”. I would use “putative inward-occluded” throughout the text.

4) The authors initially processed the nanodisc dataset without applying symmetry (C1) and noticed differences between the two Slc26a9 protomers, in which one monomer had more pronounced flexibility. A focused refinement 3D classification strategy was used to identify particles that improved the density within the region. These particles were selected and used for 3D auto-refinement with an applied C2 symmetry. Did the authors perform 3D auto-refinement at this stage without applying symmetry and did they obtain a similar map? This information should be provided in Figure 2—figure supplement 2. If this was not tried and there is a noticeable difference after auto-refinement, then the conformational differences should not be averaged by applying C2 symmetry.

5) It would be appreciated if the authors were to discuss the difference between the anion permeability and conductance series that they observed.

---

## [Author Response]

Essential revisions:1) Much of the discussion between the reviewers focused on whether there is enough evidence to establish one way or the other that SLC26A9 is a channel or a transporter. Although your structural data is compatible with a transporter model, the difference between a channel and transporter comes down to the transport rate. According to the classical definition, ion transport through a channel is diffusion limited whereas a transporter is about two to three orders of magnitude slower. Therefore, this question can only be addressed by measuring unitary fluxes. Instead, the reviewers are suggesting that the Discussion section should be reframed to provide a more balanced view of the mode of ion transport.

We have softened our claims concerning the function of SLC26A9 as transporter as opposed to a channel at several places and added an additional paragraph to the Discussion. However, we disagree with the distinction between a membrane transporter and a channel based on kinetic properties. A transporter functions by an alternate access mechanism where a saturable binding site is alternately exposed to either one or the other side of the membrane. Whereas in an alternate access mechanism, the protein needs to rearrange its conformation for each substrate molecule moved across the membrane, this is different in a channel, which, in its open conformation, provides an aqueous pore that permits diffusion of substrate down the electrochemical gradient without a change of the protein conformation. Consequently, ion channels usually operate with faster kinetics than transporters, where conformational changes confer rate limiting steps. Nevertheless, for both mechanisms, the transport rates are not uniform and span several orders of magnitude within each class reflecting the exponential relationship between the kinetics and the height of the rate-limiting barrier. Consequently, the conductance of ion channels ranges from 300 pS in case of the BK channel to low pS conductance in CLC and TMEM16 channels or sub-pS conductance in the calcium-release channel Orai. Conversely, the kinetics in transporters also bridges several orders of magnitude with fast transporters nearly approaching the rates of slow channels. One of the fastest transporters is the Cl^-^/HCO_3_^-^ exchanger SLC4A1/Band 3, whose transport domain shares a similar general protein architecture with SLC26A9 (7+7 TM inverted repeat architecture) and also transports ions by an elevator mechanism. Due to the properties of this protein architecture, with the ion binding site contained within the mobile core domain shuttling between both sides of the membrane as a rigid unit without strong interactions with the gate domain, it might be ideally suited to confer fast transport rates, especially if only small relative movements are necessary to cross an occluded barrier between intra- and extracellular sides of the membrane. Our structural data showing an inward-facing state of a transporter in the detergent structure and the movement of the core domain as rigid unit towards the outside in the nanodisc structure and the functional data where transport of ions like SCN^-^ saturates at low mM concentration (Figure 1D) are both consistent with this transport mechanism.

Ultimately, it is our view that data describing Slc26a9 is entirely consistent with a fast transporter mechanism, whereas it fails to meet all necessary criteria to be classified as a channel.

Changes to the manuscript:

“As previously shown based on whole-cell experiments (Bertrand et al., 2009; Chang et al., 2009b; Dorwart et al., 2007; Loriol et al., 2008) and confirmed here with excised patches, Slc26a9 shows large uncoupled macroscopic Cl^–^ currents that saturate only at high mM concentration. […] Additionally, our structural data clearly suggest that Slc26a9 works by an alternate-access mechanism akin to other family members.”

“Such mechanism would allow for fast transport kinetics as observed here and as previously described for the Cl^–^/HCO_3_^–^ transporter SLC4A1/Band 3, which shares a similar architecture (Arakawa et al., 2015; Brahm, 1977).”

2) The fit of the cryo-EM density to the model: I suggest the authors show the cryo-EM density for the core domain and gate domains, and show how well the structures of the transporter in the inward-open and (putative) inward-occluded states fit the density. This way readers will appreciate that the rigid body movement of the core domain does indeed occur between the two states. It is obvious to me that the rigid body motion could explain the conformational differences between the two models when I inspected the provided maps and the coordinates, which was not obvious by just reading the manuscript.

The cryo-EM densities for the core and gate sub-domains of the intermediate state with the refined models superimposed are now shown as panels C and D in Figure 2—figure supplement 4. An additional view in Panel E was added to improve the comparison of the detergent and nanodisc structures. The cryo-EM densities of the inward-facing state and the fit of the refined model is already documented in Figure 2—figure supplement 2.

Along the same line, authors need to provide more figures to show why the nanodisc reconstruction is considered as inward-occluded. Authors provided Figure 5D for comparison, but that figure just shows that there is a rigid body motion between the structures. I suggest showing surface representation like Figure 7A and B in Figure 5 or supplementary figure.

Since the nanodisc data is of low-resolution, we refrain from a detailed atomistic interpretation of its structure. Whereas our extended and updated documentation provided in Figure 2—figure supplement 4 demonstrate that the described conformational change of the core domain in the nanodisc structure compared to the detergent structure is real and displays the essential features of conformational changes during transport, a description of the structure by its molecular surface requires knowledge of sidechain-conformations to be physically meaningful. We thus replaced the term ‘inward-occluded’ conformation to ‘intermediate conformation’ throughout the manuscript, since we think that this structure shows an intermediate on the transport cycle without wanting to infer the degree of the transition. The picture of the surface of the intermediate conformation shown in Figure 8B (previously Figure 7B) was generated from the isolated domains of the nanodisc structure assuming the same sidechain conformations as observed in the detergent structure and should thus be considered as approximate model.

3) Due to the low resolution of the nanodisc-reconstituted reconstruction, I would be cautious to qualify the state as “inward-occluded”. I would use “putative inward-occluded” throughout the text.

We agree with the reviewers and have replaced the terms ‘occluded’ or ‘inward-occluded’ by ‘intermediate’ throughout the text.

4) The authors initially processed the nanodisc dataset without applying symmetry (C1) and noticed differences between the two Slc26a9 protomers, in which one monomer had more pronounced flexibility. A focused refinement 3D classification strategy was used to identify particles that improved the density within the region. These particles were selected and used for 3D auto-refinement with an applied C2 symmetry. Did the authors perform 3D auto-refinement at this stage without applying symmetry and did they obtain a similar map? This information should be provided in Figure 2—figure supplement 2. If this was not tried and there is a noticeable difference after auto-refinement, then the conformational differences should not be averaged by applying C2 symmetry.

Final 3D auto-refinement was done side-by-side with and without applying C2 symmetry and in both cases, we obtained similar maps showing two-fold symmetric features. An image of the final 3D auto-refined non-symmetrized model (6σ) at 8.4 Å was added to Figure 2—figure supplement 3E. The C1 reconstruction shows the restored density of the flexible region to almost the same extent as the C2 reconstruction. However, due to a relatively small final subset of the particles comprising the non-symmetrized model, the C1 reconstruction was only resolved to 8.4 Å. This posed a significant limitation for interpretation of possible conformational changes occurring within the transmembrane domain. In order to increase the particle pool, we decided to impose C2 symmetry, which then facilitated the generation of the model at 7.8 Å.

5) It would be appreciated if the authors were to discuss the difference between the anion permeability and conductance series that they observed.

We have added the following sentence to the Results:

“The high relative permeability of lyotropic anions shows that they face a lower energy barrier for dehydration and entry into the transport pathway compared with Cl^–^, while their lower relative conductance implies that they also encounter a deeper energy well, which in turn would decrease their overall turnover rate.”

“The lyotropic permeability sequence of Slc26a9 suggests that ions interact with a low field-strength binding site. The inability to transport SO_4_^2–^ is thus probably related to the fact that interactions within the binding site, which does not contain positive countercharges to balance the negatively-charged substrate, may not be sufficiently strong for the coordination of divalent anions.”